# Identifying Equivalent Training Dynamics

**William T. Redman**[*]
AIMdyn Inc.
UC Santa Barbara

**Juan Bello-Rivas**
Johns Hopkins University

**Maria Fonoberova**
AIMdyn Inc.

**Ryan Mohr**
AIMdyn Inc.

**Yannis G. Kevrekidis**
Johns Hopkins University

**Igor Mezić**[†]
AIMdyn Inc.
UC Santa Barbara

## Abstract

Study of the nonlinear evolution deep neural network (DNN) parameters undergo during training has uncovered regimes of distinct dynamical behavior. While a detailed understanding of these phenomena has the potential to advance improvements in training efficiency and robustness, the lack of methods for identifying when DNN models have equivalent dynamics limits the insight that can be gained from prior work. Topological conjugacy, a notion from dynamical systems theory, provides a precise definition of dynamical equivalence, offering a possible route to address this need. However, topological conjugacies have historically been challenging to compute. By leveraging advances in Koopman operator theory, we develop a framework for identifying conjugate and non-conjugate training dynamics. To validate our approach, we demonstrate that comparing Koopman eigenvalues can correctly identify a known equivalence between online mirror descent and online gradient descent. We then utilize our approach to: (a) identify non-conjugate training dynamics between shallow and wide fully connected neural networks; (b) characterize the early phase of training dynamics in convolutional neural networks; (c) uncover non-conjugate training dynamics in Transformers that do and do not undergo grokking. Our results, across a range of DNN architectures, illustrate the flexibility of our framework and highlight its potential for shedding new light on training dynamics.

## 1 Introduction

The analysis and experimentation of deep neural network (DNN) training continues to uncover new – and in some cases, surprising – phenomena. By changing the architecture, optimization hyper-parameters, and/or initialization, it is possible to identify regimes in which DNN parameters evolve along trajectories (in parameter space) with linear dynamics [1, 2], low-dimensional dynamics [3], correlated dynamics [4], lazy/rich dynamics [5, 6], and oscillatory dynamics [7, 8]. In some cases, the training dynamics have been linked with the performance of the trained model [7, 9, 10], providing new insight in DNN generalization. Additionally, detailed understanding of the dynamics has led to improvements in training efficiency [4, 11], demonstrating the practical implications such work can provide.

To obtain a more complete picture of DNN training, it is necessary to have a method by which equivalent dynamics can be identified and distinguished from other, non-equivalent dynamics. The construction of equivalence classes, which has fundamentally shaped the study of complex systems

---

[*]redmanw@aimdyn.com
[†]mezici@aimdyn.com

38th Conference on Neural Information Processing Systems (NeurIPS 2024).

in other domains (e.g., phase transitions [12], bifurcations [13], defects in materials [14]), would advance the understanding of how architecture, optimization hyper-parameters, and initialization shape DNN training and could be leveraged to search for new phenomena. However, identifying equivalent and non-equivalent training dynamics is challenged by the need for methods that:

- **Go beyond the coarseness of loss.** While useful as metrics, training and test loss can be shaped by non-dynamical features of the training (e.g., different initializations, different number of hidden units). Thus, different losses is neither necessary nor sufficient to conclude non-equivalent training dynamics.

- **Respect permutation symmetry.** DNNs are invariant to the re-ordering, within layers, of their hidden units [15]. Thus, identifying that DNN parameters evolve along trajectories that occupy distinct parts of parameter space is not sufficient to conclude non-equivalent dynamics [16, 17].

We propose to use topological conjugacy [18], a notion of dynamical equivalence developed in the field of dynamical systems theory (Sec. 3.1), to address these limitations. Historically, topological conjugacy has been difficult to compute [19], especially when the equations governing the dynamical systems under study are not known. However, recent advances in Koopman operator theory [20, 21, 22] (Sec. 3.2) have enabled the identification of topological conjugacies from data [23] (Sec. 3.3). We explore the potential of this Koopman-based approach for identifying topological conjugacies in the domain of DNN training, finding that it is able to:

- Recover a known nonlinear topological conjugacy between the training dynamics of online mirror descent and online gradient descent [24, 25, 26] (Sec. 4.1);

- Identify non-conjugate training dynamics between narrow and wide fully connected neural networks (FCNs) (Sec. 4.2);

- Demonstrate the existence of conjugate training dynamics across different random initializations of FCNs [16] (Appendix C.4);

- Characterize the early phase of training dynamics [27] in convolutional neural networks (CNNs) (Sec. 4.3);

- Uncover non-conjugate training dynamics across Transformers that do, and that do not undergo delayed generalization (i.e., "grokking") [28, 29] (Sec. 4.4).

That the same framework can be used across a number of DNN architectures to study a variety of dynamical phenomena during training demonstrates the generality of the approach. We conclude by discussing how it can be further improved to enable greater resolution of equivalent dynamics, and how it can be used to shed greater light on DNN training (Sec. 5).

## 2 Related work

### 2.1 Identification of DNN training dynamics phenomena

Analytical results have been obtained for the DNN training dynamics of shallow student-teacher [30, 31] and infinite width [1, 2] networks. For modern architectures (e.g. CNNs, Transformers), the training dynamics have been probed via analysis of computational experiments. Application of dimensionality reduction has led to the observation that parameters are quickly constrained to being optimized along low-dimensional subspaces of the high-dimensional parameter space [3, 4]. Inspection of losses, parameter and gradient magnitudes, etc. led to the identification of several transitions in the training dynamics of CNNs during the initial few epochs [27]. While insightful, this prior work cannot – except at a coarse-grained level – be used to determine whether the dynamics associated with training different DNN models (or training the same DNN model with different choices in hyper-parameters or initialization) are equivalent.

### 2.2 Koopman operator theory applied to DNN training

Data-driven implementations of Koopman operator theory have been used to model the dynamics of DNN training [32, 33, 34]. Because of the linearity of the learned Koopman models (Sec. 3.2),

using them in place of standard gradient-based methods has led to reductions in computational costs associated with DNN training. Koopman-based methods have additionally been used to meta-learn optimizers for DNNs [35, 36]. The ability of Koopman models to capture features of training dynamics has been leveraged to develop new methods for pruning DNN parameters [37, 38] and new adaptive training methods [39]. While this prior work has demonstrated that accurate Koopman operator representations of the nonlinear training dynamics can be extracted, none have utilized the theory to identify topological conjugacies.

## 3 Identifying equivalent training dynamics

### 3.1 Topological conjugacy

Given two discrete-time dynamical maps[3] $T_1 : X \to X$ and $T_2 : Y \to Y$, a natural question to ask is whether they induce equivalent dynamical behavior. There are various possibilities for defining equivalence, but dynamical systems theory has made use of the notion of **topological conjugacy** [18] to identify when a smooth invertible mapping can be used to transform trajectories of $T_1$ to those of $T_2$ (and vice versa). Formally, $T_1$ and $T_2$ are said to be topologically conjugate if there exists a homeomorphism, $h : X \to Y$, such that

$$h \circ T_1 = T_2 \circ h. \tag{1}$$

It is straightforward to identify and construct conjugacies for linear systems. Let $X = Y = \mathbb{R}^n$, and let $T_1 = A$ and $T_2 = B$, where $A, B \in \mathbb{R}^{n \times n}$. These describe linear dynamical systems, as $x(t + 1) = Ax(t)$ and $y(t + 1) = By(t)$. In this setting, $A$ and $B$ are conjugate if there exists an $H \in \mathbb{R}^{n \times n}$, such that $y(t) = Hx(t)$ and $A = H^{-1}BH$. This can happen if and only if the eigenvalues of $A$ are the same as the eigenvalues of $B$. Thus, for linear systems, topological conjugacy can be used to construct equivalence classes, partitioning the space of dynamical systems into families of matrices that have the same spectra. However, for nonlinear systems, it is challenging to prove the existence or non-existence of conjugacies [19], limiting its use as a tool. In addition, historically it has not been possible to compute topological conjugacies for systems where the underlying dynamics are not analytically known.

### 3.2 Koopman mode decomposition

Over the past two decades, Koopman operator theory has emerged as a powerful framework for studying nonlinear dynamical systems [20, 21, 22]. The Koopman operator, $U$, is an infinite dimensional linear operator that describes the time evolution of observables (i.e. functions of the underlying state-space variables, $x \in X$) that live in an appropriately defined function space, $\mathcal{F}$ (Fig. 1A). That is, the observable $g \in \mathcal{F}$ evolves as

$$Ug[x(t)] = g[Tx(t)], \tag{2}$$

where $t \in \mathbb{N}$ and $T : X \to X$ is the underlying dynamical map on the state-space $X$.

The linearity of $U$ enables a mode decomposition [termed "Koopman Mode Decomposition" (KMD)]. The KMD is similar to the mode decomposition used for linear systems analysis, except that it is defined in $\mathcal{F}$, instead of $X$. In particular, the KMD is defined as

$$U^t g(x) = \sum_{i=1}^{\infty} \lambda_i^t \phi_i(x) v_i, \tag{3}$$

where the triplet $(\lambda_i, \phi_i, v_i)$ describes the Koopman eigenvalues, eigenfunctions, and modes, respectively. If there exists a subspace $F \subset \mathcal{F}$ of finite dimension, $N \in \mathbb{N}$, that is invariant to the action of the Koopman operator, then a finite dimensional representation of the KMD can constructed,

$$U^t g(x) = \sum_{i=1}^{N} \lambda_i^t \phi_i(x) v_i. \tag{4}$$

---

[3]For the sake of space, we describe only discrete-time dynamical systems, but the theory extends to continuous time dynamical systems.

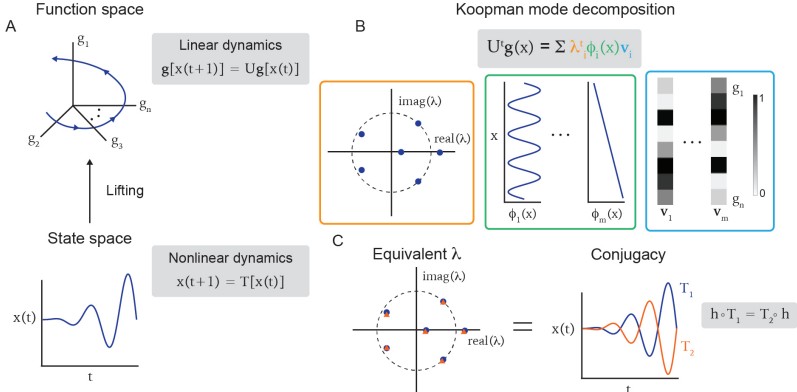

Figure 1: **Schematic of Koopman operator theory-based identification of conjugate dynamical systems.** (A) By lifting nonlinear dynamics from a finite dimensional state-space to an infinite dimensional function space, a linear representation can be achieved (from which a finite dimensional approximation can be obtained). (B) The linearity of the Koopman operator enables a mode decomposition, which includes Koopman eigenvalues (orange), eigenfunctions (green), and modes (blue). (C) Dynamical systems with the same Koopman eigenvalues are topologically conjugate.

In cases of chaotic dynamics, a representation by a finite number of Koopman modes is not achievable. Such systems are said to have continuous spectra. In order for a DNN training algorithm to be useful, it likely must avoid chaotic behavior. Therefore, we focus on training dynamics where Eq. 4 is assumed to be valid.

From Eq. 4, it can be seen that the evolution of observable functions is described as a sum of Koopman modes, each evolving at a specific time-scale (which is determined by the Koopman eigenvalues) (Fig. 1B). The Koopman eigenvalues and their associated Koopman modes and eigenfunctions can be connected to the state-space geometry of the underlying dynamical system [40].

An important feature of the Koopman eigenvalues is that they are invariant to permutations of the ordering of state-space variables. Let $x = [x_1, ..., x_n]$ and $\tilde{x} = [x_{\sigma(1)}, ..., x_{\sigma(n)}]$, where $\sigma : \{1, ..., n\} \to \{1, ..., n\}$ is a permutation and $\rho_\sigma : x \to \tilde{x}$ is the permutation mapping. That is, $\tilde{x}$ is equivalent to $x$ via a re-ordering of its labels. In this case, the action of the Koopman operator is

$$U^t \tilde{g}(\tilde{x}) = \sum_{i=1}^{N} \lambda_i^t \tilde{\phi}_i(\tilde{x}) v_i, \tag{5}$$

where $\tilde{g}(\tilde{x}) = g[\rho_\sigma^{-1}(\tilde{x})]$ and $\tilde{\phi}(\tilde{x}) = \phi[\rho_\sigma^{-1}(\tilde{x})]$. Thus, the Koopman eigenvalues are the *same* as they were for the non-permuted system. We note that the Koopman spectrum is the same for other invariances that are known to exist in DNNs, such as rescaling (of cascaded linear layers) and rotations (of query and key projections used in attention in Transformers) [41]. This makes it a generally powerful approach for studying DNN training dynamics.

While Eq. 4 is true for deterministic dynamical systems and does not hold for training via stochastic gradient descent (SGD), we believe it is still to appropriate to compute the KMD from weight trajectories for two reasons. First, theoretical work has expanded the notion of Koopman operator theory to stochastic dynamical systems [42] and defined Eq. 4 in terms of the expectation of the dynamics. This suggests that the KMD associated with SGD training will be able to inform us of the "average" dynamics during training. This will be useful to comparing different network behaviors. And second, prior work computing KMD on DNN training has found it able to sufficiently approximate the training dynamics so as to allow for the Koopman operator to be used to optimize [32, 34] and sparsify [38] DNNs. This suggests KMD can capture important aspects of the training.

Many numerical methods have been developed for approximating the KMD from data. This has enabled its successful application as a tool for spatio-temporal decomposition in providing insight into complex, real-world dynamical systems [43, 44, 45, 46, 47]. Dynamic mode decomposition (DMD) [48, 49], the most popular of these methods, has spawned many variants [50, 51, 52, 53, 54]. In general, DMD-based approaches collect $T + 1$ snapshots of data $x \in \mathbb{R}^n$, construct data matrices

$Z = [x(0), ..., x(T-1)]$ and $Z' = [x(1), ..., x(T)]$, where $Z, Z' \in \mathbb{R}^{n \times (T+1)}$, and then approximate the Koopman operator by

$$U = Z'Z^\dagger, \tag{6}$$

where $\dagger$ denotes the pseudo-inverse. Utilizing dictionaries with nonlinear functions [50] has led to improved results, demonstrating how usage of the underlying Koopman operator theory can enhance the capture of complex dynamics. In addition, leveraging Takens' Embedding Theorem [55] and using time-delayed observables has proved to be a generally powerful approach for approximating Koopman eigenvalues [44, 45, 51], an approach we make use of (Sec. 4).

## 3.3 Equivalent Koopman spectra implies topological conjugacy

Given that KMD provides a linear representation of nonlinear dynamical systems, identifying topological conjugacies through matching eigenvalues (Sec. 3.1) again becomes viable. Indeed, it has been proven that two discrete-time dynamical maps $T_1$ and $T_2$, each in the basin of attraction of a stable fixed point, are topologically conjugate if and only if the Koopman eigenvalues of the associated Koopman operators, $U_1$ and $U_2$, are the same [23] (Fig. 1C). That is, a topological conjugacy exists if and only if

$$\lambda_i^{(1)} = \lambda_i^{(2)}, \quad \forall i = 1, ..., N \tag{7}$$

where $\lambda^{(1)}$ and $\lambda^{(2)}$ correspond to the eigenvalues associated with $U_1$ and $U_2$, respectively, and $N$ is the number of Koopman modes. When the dynamical systems under study have continuous spectra, Eq. 7 does not imply topological conjugacy. As noted earlier, we do not believe this to be a major limitation when studying meaningful training dynamics. However, recent work has suggested that topological conjugacies may still be identifiable in the case of continuous spectra by using extensions of Koopman operator theory [56]. We believe this will be a fruitful direction for future work.

When the number of Koopman eigenvalues of $U_1$ is larger than the number of Koopman eigenvalues of $U_2$, the strict equivalence of Eq. 7 cannot be satisfied. However, there may exist a smooth, but non-invertible mapping $h$ from $X$ onto $Y$. In such a case, $T_1$ and $T_2$ are said to be semi-conjugate, and this can be identified when $\{\lambda_i^{(2)}\}_{i=1}^{N_2} \subset \{\lambda_j^{(1)}\}_{j=1}^{N_1}$, where $N_2 < N_1$ are the number of Koopman eigenvalues of $U_1$ and $U_2$, respectively.

Computing the KMD from data is unlikely to yield the same exact Koopman eigenvalues for conjugate dynamical systems, due to the presence of noise and finite sampling. Therefore, a method for computing the distance between eigenvalues is necessary when making comparisons. Here, we make use of the Wasserstein distance [57, 58], a metric developed in the context of optimal transport that quantifies how much one distribution must be changed to match another. This notion of "distance" is important as the Koopman eigenvalues correspond to time-scales and we expect dynamical systems with increasingly large differences between their eigenvalues will have increasingly large differences in their dynamical behavior[4]. In the case where a small, finite number of Koopman eigenvalues are computed (which can be achieved, even for systems with a large number of observables, by performing dimensionality reduction or residual based pruning of modes [53]), the Wasserstein distance can be efficiently computed by using linear sum assignment.

## 4 Results

### 4.1 Identifying conjugate optimizers

We begin by validating that numerical approximations of KMD can indeed correctly identify conjugacies in settings relevant to DNN training. To do this, we consider a recently discovered nonlinear topological conjugacy between the optimization dynamics of Online Mirror Descent (OMD) and Online Gradient Descent (OGD) [24, 25, 26] (Appendix A.1). This work has been of particular interest as OMD occurs on a convex loss landscape and OGD occurs on a non-convex loss landscape, suggesting a potential route for studying behavior of OGD in a simpler setting.

The conjugacy between OMD and OGD relies on a reparametrization of the loss function. Without prior knowledge of this reparametrization, it is challenging to identify the conjugacy by looking at only

---

[4]Note that other metrics, such as the KL-divergence, may not capture this distinction.

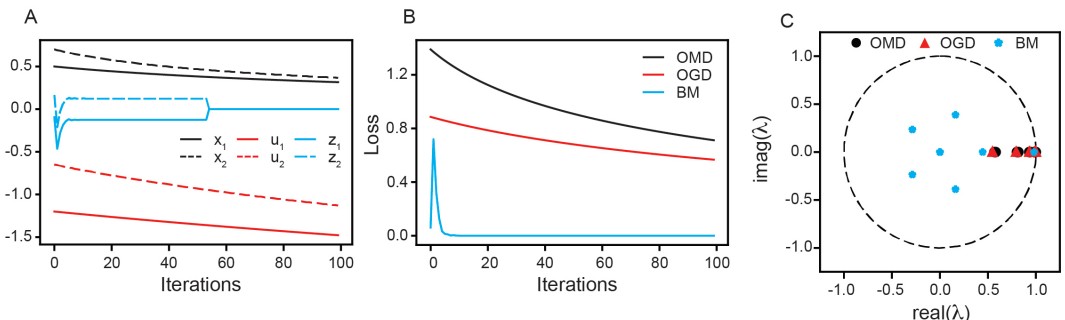

Figure 2: **Conjugacy between online mirror descent and online gradient descent is identifiable from Koopman spectra.** (A) Comparing example trajectories of variables optimized via OMD $(x_1, x_2)$, OGD $(u_1, u_2)$, and BM $(z_1, z_2)$, the existence of a conjugacy between OMD and OGD is not obvious. (B) Similarly, the existence of a conjugacy is not apparent when looking at the loss incurred by using OMD and OGD. (C) Comparing the Koopman eigenvalues associated with optimizing using OMD, OGD, and BM correctly identifies the existence of a conjugacy between OMD and OGD, and the lack of a conjugacy between OMD/OGD and BM. The function optimized is in all subfigures is $f(x) = \sum \tan(x)$.

the training trajectories or the losses (Fig. 2A, B – see Appendix A.3 for details on implementation of OGD and OMD). This highlights some of the current challenges present in identifying dynamical equivalence from data.

We compute the KMD associated with optimization using OMD and OGD by considering trajectories of both, from many initial conditions, and compare the resulting Koopman eigenvalues (Appendix A.4). We find high overlap between the two spectra (Fig. 2C, red and black dots). Additionally, the two sets of eigenvalues have the same structure. Namely, they consist only of real, positive eigenvalues. In contrast, the bisection method (BM), another optimization algorithm that is not conjugate to OMD or OGD (Appendix A.2), has associated spectra that are complex (Fig. 2C, light blue dots). Performing a randomized shuffle of the eigenvalues between algorithms (Appendix B), we find that 25% of the shuffles between OMD and OGD eigenvalues result in Wasserstein distance greater than or equal the true Wasserstein distance. This suggests the distributions are not statistically significantly distinct. However, 0% of the shuffles have Wasserstein distance greater than or equal to the true Wasserstein distance for OMD and BM, and OGD and BM, respectively. This provides evidence that the spectra of OMD/OGD and BM are statistically significantly distinct.

Similar results are obtained when applying KMD to OMD and OGD optimization of a different function (Fig. S1). Collectively, these results demonstrate that the Koopman-based spectral identification of topological conjugacies can successfully recover a known equivalence and provide support that it can be used more broadly in uncovering equivalences in DNN training dynamics.

## 4.2 Identifying the effect of width on fully connected neural network training

To start exploring the potential of our framework for identifying topological conjugacies in DNN training, we begin with a small-scale example. Namely, we consider a fully connected neural network (FCN) with only a single hidden layer, trained on MNIST (Appendix C.1). Consistent with other architectures, we find that the wider the FCN (i.e., the more units in the hidden layer), the better the performance and the lower the loss (Fig. 3A). Whether this is due to an increase in capacity, with more hidden units enabling a more refined solution, or whether this is due to a change in the training dynamics, leading to a better traversal of the loss landscape, is – at this point – unclear.

Computing the Koopman eigenvalues associated with training FCNs of varying width (Fig. 3D – see Appendix C.2 for details), we find that narrow ($h = 5$) and wide ($h = 40$) FCNs have training dynamics that are non-conjugate, as their Koopman spectra are non-overlapping (Fig. 3E, F). This suggests that the training dynamics undergo a fundamental change as width increases. However, for FCNs with intermediate width ($h = 10$), the training dynamics are more aligned with the wide FCNs (Fig. 3E, F), suggesting conjugate dynamical behavior. The dynamical difference in training narrow and wide FCNs is also supported by performing the eigenvalue shuffle analysis (Appendix B).

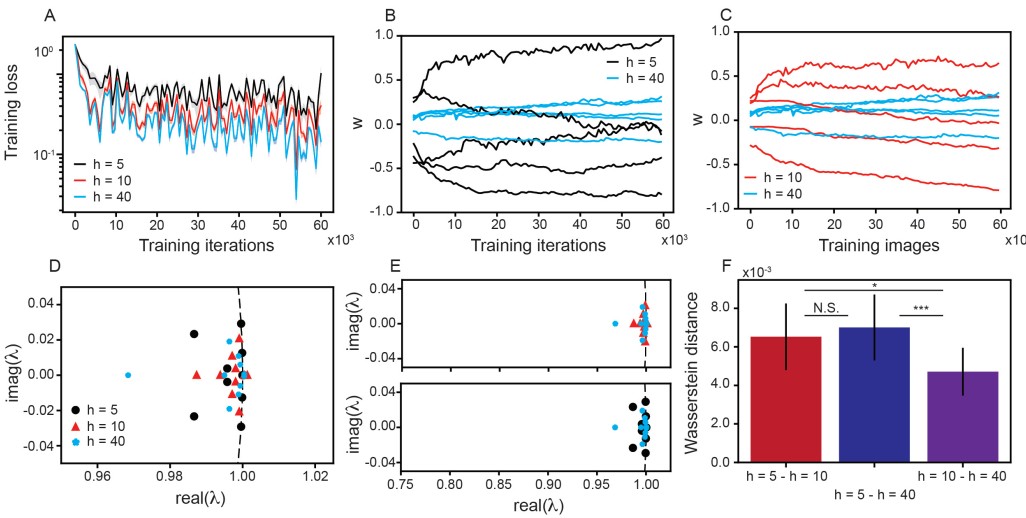

Figure 3: **Narrow and wide fully connected neural networks have non-conjugate training dynamics.** (A) Training loss curves for FCNs with hidden layer widths $h = 5, 10$, and $40$. Solid line is mean and shaded area is $\pm$ standard deviation across 25 independently trained networks. (B), (C) Example weight trajectories, across training iterations, for narrow, intermediate, and wide FCNs. (D) Koopman eigenvalues associated with training FCNs of varying width. (E) Same as (D), but zoomed out and with the eigenvalues associated with $h = 5$ and $h = 10$ compared to those associated with $h = 40$. Dashed line in (D) and (E) denotes unit circle. (F) Wasserstein distance between Koopman eigenvalues associated with training FCNs of varying width. Error bars are $\pm$ standard deviation across 25 independently trained FCNs. Kolmogorov–Smirnov (KS) tests were performed to assess statistical significance of distance: $*$ denotes $p < 0.01$ and $* * *$ denotes $p < 0.0001$.

In particular, a much larger number of the shuffles between $h = 10$ and $h = 40$ eigenvalues have Wasserstein distance greater than or equal to the true Wasserstein distance than between $h = 5$ and $h = 40$ (81% vs. 55%), although significance is not reached. Similar results were found when using the GeLU instead of ReLU activations [59] (Fig. S3), demonstrating that our results are consistent across FCNs with similar activation functions. Thus, we conclude that the additional improvement in performance observed when increasing the network width from $h = 10$ to $h = 40$ comes more from an increase in capacity, than from a change in training dynamics. Identifying this was not possible by solely comparing the loss or weights (Fig. 3A–C), demonstrating the advantage of the Koopman-based approach for identifying equivalent and non-equivalent DNN training dynamics.

To further study the behavior of FCN training dynamics, we also compared the computed Koopman spectra of $h = 40$ networks trained from different random initial conditions (Appendix C.4). Prior work has proven that different random initializations of sufficiently wide FCNs converge to local minima that have no loss barrier along the linear interpolation between them, when taking into account permutation symmetry [16]. This suggests conjugate training dynamics, although this has not been explicitly shown. In support of this hypothesis, we find examples of FCNs, trained from different random initializations, with nearly identical Koopman spectra (Fig. S4A).

### 4.3 Identifying dynamical transitions in convolutional neural network training

Prior work has argued that CNNs undergo transitions in their training dynamics during the early part of training (i.e. the first several epochs), and that these transitions are similar across different CNN architectures [27]. However, dynamical systems based methods were not used for analysis. Instead, this observation relied on coarse-grained observables (e.g., training loss, magnitude of gradients) to define the transitions and to determine when they occur.

To understand whether such results hold when considering the training dynamics at a finer-scale, we utilize our Koopman-based framework. To do this, we split the first epoch of training into windows of 100 training iterations. We compute the Koopman eigenvalues associated with dynamics that occur in each window and denote them by $\lambda_{t_1:t_2}$, where $t_1 < t_2$ are the first and last training iteration in the window. We then measure the Wasserstein distance between all combinations of pairs of eigenvalues. This enables us to quantitatively assess transient dynamical behavior and identify when

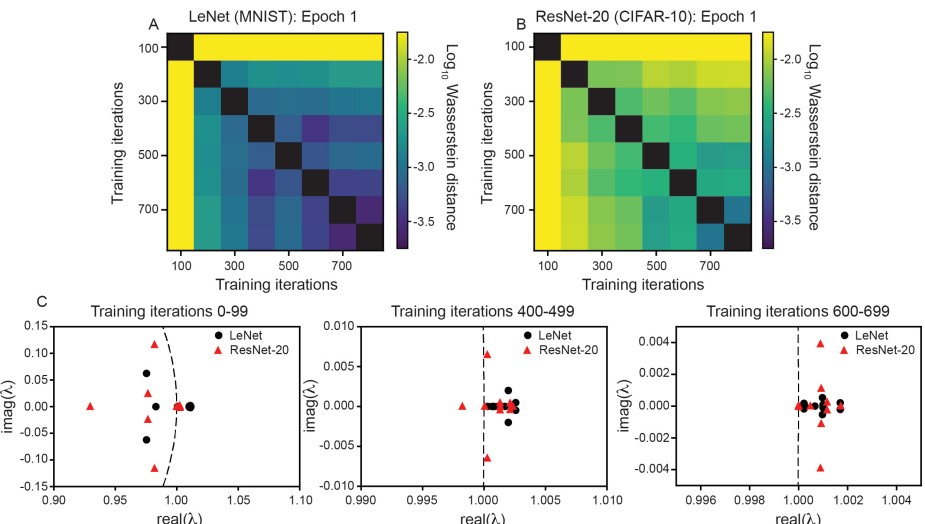

Figure 4: **Koopman-based framework enables identification of transitions in dynamics during the early phase of training for LeNet and ResNet-20.** (A) $\text{Log}_{10}$ Wasserstein distance between Koopman eigenvalues associated with LeNet training over windows of 100 training iterations during epoch 1. (B) Same as (A), but for ResNet-20 training. (C) Koopman eigenvalues associated with the dynamics that occur during training iterations intervals 0–99, 400–499, and 600–699. Dashed line denotes the unit circle.

in the early phase of training the dynamics transition from one equivalence class to another. We apply our approach to LeNet [60], a simple CNN trained on MNIST, and ResNet-20 [61], trained on CIFAR-10 (see Appendix D.1 for details).

We find that, for both LeNet and ResNet-20, the first 100 training iterations have the most distinct Koopman eigenvalues, as the Wasserstein distance between $\lambda_{0:99}$ and all other eigenvalues is large (Fig. 4A, B – bright yellow first column and row). In addition, for both LeNet and ResNet-20, the training dynamics become similar after 700 training iterations, as the Wasserstein distance between $\lambda_{600:699}$ and $\lambda_{700:799}$ is small (Fig. 4A, B – dark blue square around diagonal in lower righthand corner). This is in agreement with the timeline found by Frankle et al. (2020) [27]. However, we additionally find that the dynamics that occur between 100 and 700 training iterations exhibit greater change for ResNet-20 than for LeNet, as there is a larger Wasserstein distance between Koopman eigenvalues. This suggests a difference in dynamical behavior between the architectures. By examining the Koopman eigenvalues associated with different training iteration windows, we find non-overlapping spectra (Fig. 4C). Performing the eigenvalue shuffle analysis (Appendix B), we find evidence that the first 4 splits of 100 training steps have statistically significant different associated Koopman eigenvalues, as the 2%, 2%, 0%, and 4% of the shuffles had Koopman eigenvalues greater than or equal to the true Wasserstein distance. This suggests a lack of topological conjugacy between the earliest training dynamics of ResNet-20 and LeNet, despite the fact that the general timeline in transitions in dynamics is similar between the architectures.

To understand how the training dynamics change over a larger span of training time, we perform the same analysis, but computing the Koopman eigenvalues from the dynamics that occur during each epoch (Appendix D.3). We find that, at this coarser grain scale, both architectures see a similar evolution of their training dynamics. In particular, we find that the first epoch has the most distinct dynamics (Fig. S5A, B – yellow first column and row), and the subsequent epochs have dynamics that become increasingly more similar (Fig. S5A, B – increasing size of dark blue blocks centered on the diagonal).

Taken together, our Koopman-based analysis supports prior decomposition of the early phase of CNN training dynamics into regimes separated by transitions that occur after a similar number of training iterations across architectures [27]. However, with a finer-scale resolution of the dynamics, we additionally find that LeNet and ResNet-20 have non-conjugate training, demonstrating that the exact training dynamics are architecture-specific.

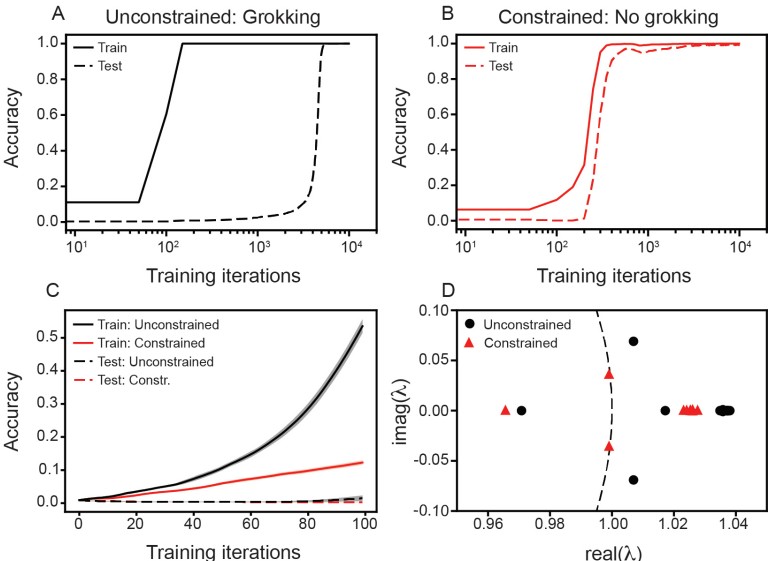

Figure 5: **Transformers that do, and that do not undergo grokking have early training dynamics that are not conjugate.** (A) Train and test loss, as a function of training steps, for a Transformer model that undergoes grokking [28]. (B) Same as (A), but for a Transformer whose training is constrained to have a constant weight norm [62]. In this case, no grokking is observed. (C) In the first 100 training steps, little difference is seen between the test loss of Transformers with and without constrained training. Lines are mean and shaded area is ± standard deviation across 20 independently trained networks. (D) Koopman eigenvalues associated with the dynamics that occur over the first 100 training iterations for Transformers that do, and that not undergo grokking.

## 4.4 Identifying non-conjugate training dynamics for Transformers that do and do not grok

Since the discovery that Transformers trained on algorithmic data (e.g., modular addition) undergo delayed generalization ("grokking" – Fig. 5A) [28], considerable effort has been invested to understand how this arises. One particularly influential theory is that the norm of the weights at initialization plays a large role. In particular, it was shown that single layer Transformers, initialized at weights with a sufficiently small norm, have training and test loss landscapes that are "aligned", while the same single layer Transformers, initialized at weights with a sufficiently large norm, have training and test loss landscapes that are "mis-aligned" [62]. Constraining the norm of the weights to be small prevents grokking, with train and test accuracy increasing at similar training iterations (Fig. 5B) [62].

What role the training dynamics play in grokking remains less understood. In particular, the extent to which constraining the weight norm changes the training dynamics (which could shed additional light on grokking) has yet to be explored. We therefore compute the Koopman eigenvalues associated with the training of constrained and unconstrained Transformers on modular addition (Appendix E). We use the dynamics from the earliest part of training, namely the first 100 training iterations (Fig. 5C). We do this to avoid trivially seeing a difference, given the small weight changes that Transformers which undergo grokking make when the training accuracy is high.

We find that the Koopman eigenvalues are distinct (Fig. 5D). In addition to a gap between the computed eigenvalues, we find that the dynamics associated with training the constrained Transformer has a pair of complex conjugate eigenvalues that lie along the unit circle, whereas the dynamics associated with training the unconstrained Transformer has a pair of complex conjugate eigenvalues outside of the unit circle. This suggests a difference in stability properties, as Koopman eigenvalues with magnitude greater than 1 (i.e. those that lie outside the unit circle) correspond to unstable dynamics. Similar results were found when computing the Koopman eigenvalues associated with the training of the unconstrained Transformer over a longer training time window (Fig. S6).

These results suggest a non-conjugacy in the training dynamics of Transformers that do, and those that do not undergo grokking. In particular, constraining the weight norm appears to lead to more stable training dynamics, which may be due to the selection of a better subnetwork to train [63]. Additionally, these results suggest that it may be possible to identify grokking before it happens [64].

# 5 Discussion

Motivated by the need for quantitative methods that can determine when the training dynamics of DNN parameter trajectories are equivalent, we utilized advances in Koopman operator theory to develop a framework that can identify topological conjugacies. This Koopman based identification of conjugate training dynamics is invariant to permutations of DNN parameters (Eq. 5), a necessary feature for methods used to study DNN training dynamics [15, 16, 17]. By applying our approach to the dynamics associated with optimization using OMD and OGD, we were able to validate that numerical implementations of KMD can identify conjugacies which are known to exist [24, 25, 26] (Fig. 2C). Additionally, this example demonstrates challenges existing approaches for comparing DNN training dynamics face, as comparing the losses and the parameter evolutions of OMD and OGD does not lead to a clear indication of the underlying equivalence (Fig. 2A, B).

Leveraging our Koopman-based approach on the training dynamics of DNNs of varying architectures led to several insights. First, we found evidence that shallow and wide FCNs have non-conjugate training dynamics (Fig. 3). This is consistent with theoretical and experimental work showing that FCN width can lead to lazy and rich training regimes [6]. This provides further evidence that our Koopman-based approach can correctly identify equivalent and non-equivalent training dynamics. In addition, we find that FCNs of intermediate width have Koopman eigenvalues that are more similar to those of wide FCNs (Fig. 3), demonstrating that our approach can provide insight beyond the wide and shallow regimes. Second, applying our framework to the dynamics of CNNs, we found transitions in the dynamics during the early phase of training, consistent with prior work [27] (Fig. 4). However, by closely examining the Koopman eigenvalues, we found non-conjugate dynamics between different CNN architectures, suggesting fundamental differences in training. These distinct dynamical features are aligned with previous observations of different behaviors when training sparse CNNs [11, 38]. And third, we found that Transformers that do, and that do not undergo grokking have non-conjugate training dynamics (Fig. 5). By focusing on the early phase of Transformer training, we avoid trivially finding this due to differences in the training loss. Additionally, this provides evidence for the ability to anticipate grokking before it happens [64].

Our framework is similar in spirit to an approach that categorizes iterative algorithms from a controlled dynamical systems perspective [65, 66]. However, such an approach requires access to the underlying equations to identify equivalence classes, which our data-driven, Koopman based framework does not [67, 68]. Work concurrent to ours has leveraged a similar approach to study the dynamics of recurrent neural networks [69]. However, Ostrow et al. (2023) [69] studied the dynamics of the activations and not the dynamics of network parameters, which is the focus of this paper.

**Limitations.** Numerical implementations that compute the KMD are only approximations to the action of the true Koopman operator. As such, they are subject to the same difficulties as other data-driven approaches. These include the selection of hyper-parameters associated with the construction of the Koopman operator, the choice of observable functions, and the number of modes considered. To mitigate the effect these limitations might have on our analysis, we used DMD-RRR, a state-of-the-art numerical approach for KMD [53], and time-delayed observables, which have been found to provide robust results across a range of domains [44, 45, 51]. Determining the existence of a topological conjugacy between two dynamical systems requires assessing whether their associated Koopman eigenvalues are sufficiently similar. While in some cases this is clear (e.g. identical Koopman eigenvalues associated with optimization using OMD and OGD – Fig. 2, distinct Koopman eigenvalues associated with training LeNet and ResNet-20 – Fig. 4), in other cases it is less apparent. To quantify these differences, we made use of the Wasserstein distance and attempted to compute significance with a randomized shuffle control. While a natural choice, additional work remains to connect the magnitude of the Wasserstein distance to the divergence of the dynamical properties associated with training DNN models.

**Future directions.** The ability of our Koopman-based approach to identify conjugacies between iterative optimization algorithms suggests its potential for data-driven discovery and generation of new classes of algorithms [70, 71, 72]. By identifying equivalent training dynamics of DNNs, it may be possible to use our approach for learning mappings that transform one DNN model to another [73]. Finally, the characterization of the Koopman eigenvalues associated with training a wide range of DNN models varying in architecture, optimization hyper-parameters, and initialization will enable a detailed understanding of how these properties shape DNN training. Leveraging this understanding may lead to improved methods for DNN training and model development.

## Acknowledgments

We thank members of AIMdyn Inc., Mitchell Ostrow, Adam Eisen, Ila Fiete, and members of the Fiete Group for helpful discussion surrounding this work. We thank the anonymous NeurIPS reviewers for their thorough and detailed feedback, which strengthened or work. This material is based upon work supported by the Air Force Office of Scientific Research under award number FA9550-22-1-0531.

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

# A Online mirror and online gradient descent

## A.1 Conjugacy between OMD and OGD

Here we outline the conjugacy between Online Mirror Descent (OMD) and Online Gradient Descent (OGD). We follow the notation and framing presented in Ghai et al. (2022) [26].

Let $\mathcal{K}$ be a convex set. OMD is applied to find a minimum of the function $f$ on $\mathcal{K}$, subject to a convex regularizer $R$. For each iteration of OMD, the state of the algorithm (initialized at $x(0) \in \mathcal{K}$), is updated by performing $(\nabla R)^{-1}(\nabla R[x(t)] - \eta \nabla f[x(t)])$, where $\eta$ is the learning rate. Because this step may not be in $\mathcal{K}$, the Bregman projection operator, $\Pi_\mathcal{K}^R(x) = \arg\min_{y \in \mathcal{K}} D_R(y||x)$, is used. OGD, on the other hand, is applied to a (possibly) non-convex set $\mathcal{K}'$ and a (possibly) non-convex function $\tilde{f}$. As with OMD, each iteration of OGD involves updating the state (initialized at $u(0) \in \mathcal{K}'$) and projecting the update back into the set $\mathcal{K}'$. In this case, the update is $u(t) - \eta \nabla \tilde{f}[u(t)]$, where again $\eta$ is the learning rate and the projection is done via the Euclidean projection operator, $\Pi_{\mathcal{K}'}(x)$ (see Algorithms 1 and 2 for pseudocode implementations of both algorithms). When $\nabla \tilde{f}[u(t)] = \nabla f(q[u(t)])$ and $\mathcal{K}' = q^{-1}(\mathcal{K})$, the outputs of the two algorithms are equivalent via the mapping $q$ (i.e., $q$ is the topological conjugacy or reparameterization). A key theorem of [25] showed that, in continuous-time, if $x(t) = q([u(t)])$, then $\frac{\partial u}{\partial t} = -\eta \nabla f(q[u(t)])$ [26].

---

**Algorithm 1** Online Mirror Descent [26]

---

0: **Input:** $x(0) \in \mathcal{K}, R, \eta, f$
0: **for** $t = 0, ..., T - 1$ **do**
0:     $y(t+1) = (\nabla R)^{-1}(\nabla R[x(t)] - \eta \nabla f[x(t)])$
0:     $x(t+1) = \Pi_\mathcal{K}^R[y(t+1)]$

---

**Algorithm 2** Online Gradient Descent [26]

---

0: **Input:** $u(0) \in \mathcal{K}', \eta, \tilde{f}$
0: **for** $t = 0, ..., T - 1$ **do**
0:     $v(t+1) = u(t) - \eta \nabla \tilde{f}[u(t)]$
0:     $u(t+1) = \Pi_{\mathcal{K}'}[v(t+1)]$

---

## A.2 Bisection method

Let $\mathcal{K}'' = [c, d]^d$ where $f(c) < 0$, $f(d) > 0$, and there exists only one $z^* \in [c, d]^d$ s.t $f(z^*) = 0^d$. Let $a(0), b(0) \in \mathcal{K}''$ s.t. $f[a(0)] < 0$ and $f[b(0)] > 0$. Define $z(t) = [a(t) + b(t)]/2$. For each iteration of the Bisection Method (BM), if $f[z(t)] < 0$, then $[a(t), b(t)]$ is updated to $[z(t), b(t)]$. Otherwise $[a(t), b(t)]$ is updated to $[a(t), z(t)]$ (see Algorithm 3 for pseudocode implementation).

Because not one but three variables are being updated at each iteration of the BM, $[a(t), b(t), z(t)]$, the BM can exhibit several distinct properties from OMD and OGD. First, how much the updated $z(t + 1)$ differs from $z(t)$ depends on $a(t)$ and $b(t)$. Thus, $||z(t + 1) - z(t)||_2$ can be much larger than the steps OMD and OGD takes. This global property enables it to escape local minima that OMD/OGD get stuck in, but can also lead to increases in loss. This behavior can be seen in Fig. 2B. Second, because $f[a(t)] < 0 < f[b(t)]$ and because $z(t + 1)$ is either $a(t)$ or $b(t)$, the outputs of the BM [i.e., $z(t)$] can flip sign. This kind of oscillatory behavior can be seen in Fig. 2A. This would not happen with OMD or OGD, assuming a sufficiently small learning rate. For these reasons, we expect that the bisection method is *not* conjugate to OMD or OGD. The distinct Koopman spectra (Fig. 2C) demonstrate that our framework properly identifies this non-conjugacy.

---

**Algorithm 3** Bisection Method

---

0: **Input:** $f, a(0) \in \mathcal{K}'', b(0) \in \mathcal{K}''$ s.t. $f[a(0)] < 0 < f[a(b)]$
0: **for** $t = 0, ..., T - 1$ **do**
0:     $z(t) = [a(t) + b(t)]/2$
0:     **if** $f[z(t)] < 0$ **then**
0:         $a(t + 1) = z(t)$
0:         $b(t + 1) = b(t)$
0:     **else**
0:         $a(t + 1) = a(t)$
0:         $b(t + 1) = z(t)$

---

## A.3 Numerical experiments

To validate that the numerically computed Koopman eigenvalues, corresponding to OMD and OGD applied to specific problems, encode sufficient information to correctly identify the conjugacy, we chose to test them on the example of log barrier regularization with exponential reparameterization, as presented in Ghai et al. (2022) [26]. In particular, we set $R = -\sum_{i=1}^d \log(x_i)$, $\mathcal{K} = [0.01, 1.0]^d$, $\mathcal{K}' = [-4.6, 0.0]^d$, and $\mathcal{K}'' = [-4/3, 8/7]^d$ (with $d = 2$), and $q(u) = \exp(u)$. This is an example of a nonlinear conjugacy.

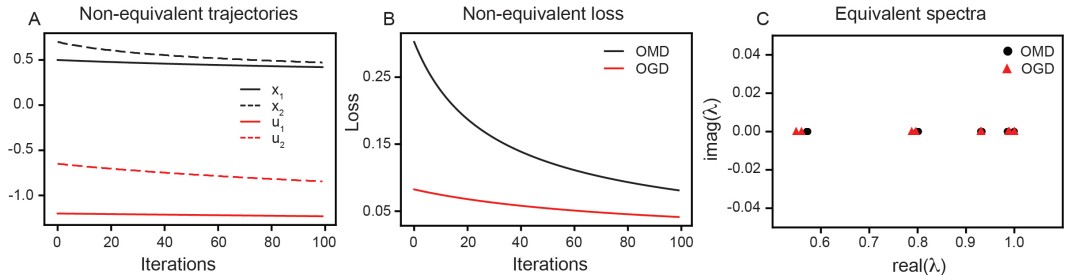

Figure S1: **Conjugacy between online mirror descent and online gradient descent is identifiable from Koopman spectra.** (A) Comparing example trajectories of variables optimized via OMD ($x_1, x_2$) and OGD ($u_1, u_2$), the existence of a conjugacy is not obvious. (B) Similarly, the existence of a conjugacy is not apparent when looking at the loss incurred by using OMD and OGD. (C) Comparing the Koopman eigenvalues associated with optimizing using OMD and OGD correctly identifies the existence of a conjugacy. The function optimized is in all subfigures is $f(x) = \sum x^4$. Note that the BM was not used in this figure as $f(x)$ is symmetric around its minimum, which does not satisfy the assumption that $f(a) < 0$.

We applied OMD and OGD on $f = \sum_{i=1}^{d} x_i^4$ (Fig. S1) and $f = \sum_{i=1}^{d} \tan(x_i)$ (Fig. 2), for $T = 100$ time steps, with a learning rate of $\eta = 0.01$. We evolved 25 initial conditions, with $x(0) \in \{0.1, 0.3, 0.5, 0.7, 0.9\} \times \{0.1, 0.3, 0.5, 0.7, 0.9\}$ and $u(0) \in \{-2.30, -1.75, -1.20, -0.65, -0.10\} \times \{-2.30, -1.75, -1.20, -0.65, -0.10\}$. When we used the BM (Fig. 2), we sampled 25 initial conditions, with $a(0) \in \{-16/12, -13/12, -10/12, -7/12, -4/12\} \times \{-16/12, -13/12, -10/12, -7/12, -4/12\}$ and $b(0) \in \{1/7, 0.393, 0.643, 0.893, 8/7\} \times \{1/7, 0.393, 0.643, 0.893, 8/7\}$. Note that, for simplicity, we consider each $a(0)$ and $b(0)$ only once (leaving 25 initial conditions). Using the resulting trajectories, we computed the KMD (see Appendix A.4). All experiments were run on a MacBook Air with an Apple M1 chip, 1 CPU, and no GPUs. Code implementing our experiments can be found at
`https://github.com/william-redman/Identifying_Equivalent_Training_Dynamics`.

### A.4 Computing the Koopman mode decomposition

To compute the KMD associated with optimization using OMD and OGD on $f(x) = \sum_{i=1}^{d} \tan(x_i)$ (Fig. 2) and $f(x) = \sum_{i=1}^{d} x_i^4$ (Fig. S1), we saved the values of $x(t)$ and $u(t)$ across the $T = 100$ training steps. These were concatenated into tensors $X, U \in \mathbb{R}^{2 \times 100 \times 25}$. Four time-delays [44, 45, 51] were applied, and the resulting tensors were flattened along the last dimension. This led to matrices $Z_X, Z_U \in \mathbb{R}^{10 \times 2375}$. We applied DMD-RRR [53] on these matrices to compute the Koopman eigenvalues. The same approach was used to compute the KMD associated with optimization via BM.

## B   Randomized shuffle control

While the Wasserstein distance gives a natural way to quantify how similar the Koopman spectra associated with two dynamical systems are, it remains an open question on how to best interpret the magnitude of the computed distance. In particular, if $\Lambda^{(1)} = \{\lambda_1^{(1)}, ..., \lambda_N^{(1)}\}$ and $\Lambda^{(2)} = \{\lambda_1^{(2)}, ..., \lambda_N^{(2)}\}$, what value of the Wasserstein distance $\omega = W_2\left(\Lambda^{(1)}, \Lambda^{(2)}\right)$ is "sufficiently small" enough that we can confidently conclude the two dynamical systems are topologically conjugate?

To help increase the transparency and interpretability of our results, we develop a randomized shuffle control to act as a baseline to assess how significantly distinct $\Lambda^{(1)}$ and $\Lambda^{(2)}$ are. In particular, by comparing $\omega$ to a distribution of Wasserstein distances computed from randomly shuffling $\Lambda^{(1)}$ and $\Lambda^{(2)}$, we may estimate whether the true Koopman eigenvalues are more or less "distant" than would be expected by chance.

One challenge that emerges in creating a good randomized shuffle control is the fact that the Koopman eigenvalues correspond to dynamical properties of the underlying system (e.g., decay rates, oscillations, fixed points). Therefore, if we take the naive approach and randomly assign half of $\Lambda = \{\Lambda^{(1)}, \Lambda^{(2)}\}$ to $\Lambda'^{(1)}$ and the other half to $\Lambda'^{(2)}$, then $\Lambda'^{(1)}$ and $\Lambda'^{(2)}$ could be "non-natural", having only slow or fast decays modes, no fixed points, etc. In such a case, we expect $\omega < \omega' = W_2\left(\Lambda^{(1)}, \Lambda^{(2)}\right)$ in general. This makes the naive approach a poor baseline.

To attempt to construct a stronger and more informative baseline, we randomly assign eigenvalues to $\Lambda'^{(1)}$ and $\Lambda'^{(2)}$ in the following manner. First, by computing $W_2\left(\Lambda^{(1)}, \Lambda^{(2)}\right)$ we identify the "assignment" $\sigma$ : $\{1, ..., N\} \rightarrow \{1, ..., N\}$ that maps the order of $\Lambda^{(2)}$ to be as close to $\Lambda^{(1)}$ as possible. In particular, $\sigma$ is defined as

$$\min_{\sigma} \sum_{i=1}^{N} ||\lambda_i^{(1)} - \lambda_{\sigma(i)}^{(1)}||_2. \tag{8}$$

To generate the shuffled eigenvalues, we then place $\lambda_i^{(1)}$ in $\Lambda'^{(1)}$ with 50% probability and in $\Lambda'^{(2)}$ with 50% probability. The eigenvalue in $\Lambda^{(2)}$ that is closest to $\lambda_i^{(1)}$ via the assignment $\sigma$ [i.e., $\lambda_{\sigma(i)}^{(2)}$] is then placed in whichever $\Lambda'^{(1)}$ or $\Lambda'^{(2)}$ that $\lambda_i^{(1)}$ is not. Performing $n_{\text{shuff}} = 100$ shuffles, we get a distribution of Wasserstein distances between the shuffled eigenvalues, $\{\omega_1', ..., \omega_{n_{\text{shuff}}}'\}$. We report the number of shuffles that have a Wasserstein distance greater than or equal to $\omega$. In the case where the two sets of Koopman eigenvalues are highly distinct, we expect there to be few shuffles that satisfy $\omega' \geq \omega$. In contrast, when the Koopman eigenvalues are highly overlapping, we expect there to be more shuffles that satisfy $\omega' \geq \omega$. For this reason, we report the percent of shuffles with $\omega' \geq \omega$ in the main text.

This approach to defining the randomized shuffle control has two useful properties. First, $\Lambda'^{(1)}$ and $\Lambda'^{(2)}$ have as similar magnitudes of eigenvalues to $\Lambda^{(1)}$ and $\Lambda^{(2)}$ as possible. And second, if $\lambda_i^{(1)}$ and $\lambda_{i+1}^{(1)}$ are complex conjugate pair of eigenvalues, whose closest matches are $\lambda_{\sigma(i)}^{(2)}$ and $\lambda_{\sigma(i+1)}^{(2)}$ that are real only, then the complex conjugate pair can be "split" when creating $\Lambda'^{(1)}$ and $\Lambda'^{(2)}$. This can lead to $\omega' < \omega$. Thus, there is a "penalty" when the number of complex conjugate eigenvalues does not match between $\Lambda^{(1)}$ and $\Lambda^{(2)}$.

We note that this randomized shuffle control is not a perfect construct and represents one possible way of generating significance. We hope that future work will enable rigorous comparison between Koopman eigenvalues by developing computation schemes for estimating how much two dynamical systems diverge given the differences in their approximate spectra.

## C   Fully connected neural networks

### C.1   Training experiment details

FCNs with only a single hidden layer are trained on MNIST, for one epoch, using SGD. Training was performed using PyTorch. All hyper-parameters used for training are presented in Table S1. All experiments were run on a MacBook Air with an Apple M1 chip, 1 CPU, and no GPUs. Code implementing our experiments can be found at `https://github.com/william-redman/Identifying_Equivalent_Training_Dynamics`.

| Hyper-parameters | Values |
|---|---|
| Learning rate ($\eta$) | 0.1 |
| Batch size ($b$) | 60 |
| Optimizer | SGD |
| Epochs | 1 |
| Activation function | ReLU |

Table S1: **Hyper-parameters used for FCN training in Sec. 4.2.**

### C.2   Computing the Koopman mode decomposition

A challenge in computing the KMD associated with training FCNs is that, in order to accurately approximate the Koopman eigenvalues, multiple trajectories must be sampled. However, given that FCNs (and other DNN models) can have loss landscapes with multiple local minima, training from different random initializations can lead to different trajectories with different dynamical properties. To address this, we perform the following three steps:

- We randomly sample an initialization for the network input and output weights. We use the standard PyTorch initialization scheme, with weights being uniformly sampled in $[-\sqrt{k}, \sqrt{k}]$, where $k$ is the number of input features. We denote this initialization by $W_0$. We then train the FCN, from $W_0$, for a full epoch.

- We sample another $n_s - 1$ initializations of the FCN, for $n_s \in \mathbb{N}$. Instead of randomly sampling a new set of parameters, we consider a perturbation of $W_0$. Namely, $W_0[1 + \varepsilon\mathcal{N}(0, 1)]$, where $\mathcal{N}(0, 1)$ is a Gaussian distribution with zero mean and unit variance. The FCNs were then trained

from these initializations, using the same batch order as the one used to train the FCN from $W_0$. In our experiments, we set $n_s = 10$ and $\varepsilon = 0.001$. To investigate whether training from the perturbed initialization did indeed lead to dynamics that were in the same basin of attraction as training from $W_0$, we computed the ratio of their end test loss with the end test loss of the network initialized at $W_0$. We find that the ratio is centered around 1 (Fig. S2), providing evidence for that the training dynamics are restricted to the basin of attraction of the same local minimum.

- We repeated the steps above $n_n - 1$ times, for $n_n \in \mathbb{N}$. The weight evolutions, from each set of networks, were saved separately and the KMD was computed on each one independently. In our experiments, we set $n_n = 25$.

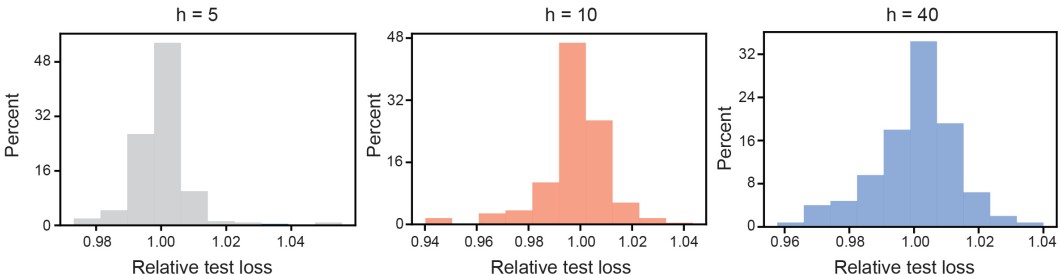

Figure S2: **Perturbing network weights leads to similar loss relative to original model.** To evaluate whether training the FCN from the perturbed initialization led to trajectories that lay within the same basin of attraction as the unperturbed initialization, we compute the relative test loss (test loss for perturbed initialization divided by test loss for unperturbed initialization). For all FCN widths, we see that the distribution of relative test loss is peaked at 1, demonstrating that the perturbed FCNs converge to a similar test loss as the unperturbed initialization. This provides evidence that the trajectories lie in the same local minimum basin.

To compute the KMD, we considered as observables the weights from the hidden layer to the output layer. This choice was made because: 1) the values of these weights determine the weight evolution of the earlier layers (due to backprop); 2) there are fewer weights from the hidden layer to the output layer, than there are from the input layer to the hidden layer, enabling our approach to be more computationally tractable. To enrich our observables, we considered time-delays [44, 45, 51]. Because we considered FCNs of differing width, using the same number of time-delays leads to matrices of different dimension. Therefore, we fixed the ratio of $d$ (number of time-delays) to $h$ (the number of units in the hidden layer), setting $d = 64$ for FCNs with $h = 5$, $d = 32$ for FCNs with $h = 10$, and $d = 8$ for FCNs with $h = 40$.

We applied DMD-RRR [53] on these time-delayed observables. When principal component analysis (PCA) was performed on these observables, it was found that a lower-dimensional subspace contained a large percentage of the variance. This is in-line with previous work finding the weights of DNNs traverse low dimensional subspaces [3]. Therefore, we considered only the top 10 Koopman modes, using a reduced singular value decomposition (SVD).

## C.3 GeLU FCNs

To understand how robust our conclusion that shallow and wide FCNs have non-conjugate training dynamics is, we perform the same Koopman-based analysis on FCNs with GeLU activation functions [59]. Given the similarity between ReLU and GeLU, we expect that they should have similar training dynamics behavior. Computing the Koopman eigenvalues and comparing across FCNs of varying widths, we find very similar results (compare Fig. S3 with Fig. 3). This demonstrates that our Koopman based framework is robust to (minor) hyper-parameter differences. A full examination of how other choices of activation function (especially those that introduce "squashing" – e.g., sigmoid) impact the training dynamics would be an interesting future direction to pursue.

## C.4 Conjugate training dynamics across random initializations of FCNs

Motivated by recent work arguing that different random initializations of DNNs converge to solutions with no loss barrier in the linear interpolation between them [16], when taking into account the inherent permutation symmetry of the parameters, we asked whether our Koopman-based framework identified conjugate training dynamics for different initialized FCNs. We examine how similar the Koopman spectra for all 25 independently trained FCNs of width $h = 40$ are. We chose this because Entezari et al. (2022) [16] proved their result for sufficiently wide FCNs, but sufficiently narrow FCNs exhibited a loss barrier along the linear interpolation, when

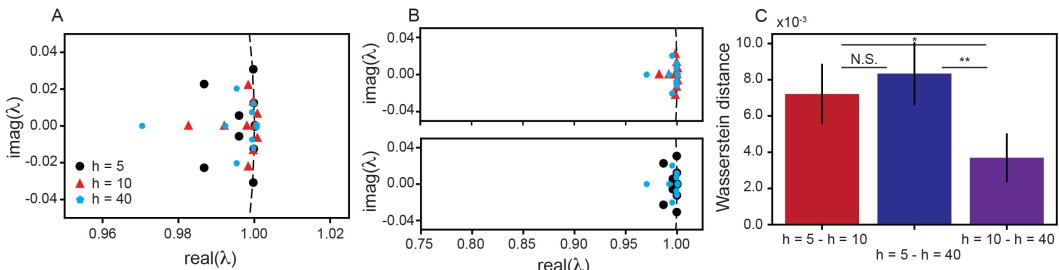

Figure S3: **Narrow and wide fully connected neural networks have non-conjugate training dynamics when using GeLU activation functions.** Same as Fig. 3D–F in the main text, but for FCNs using GeLU activation functions. Error bars are $\pm$ standard deviation across 10 independently trained FCNs.

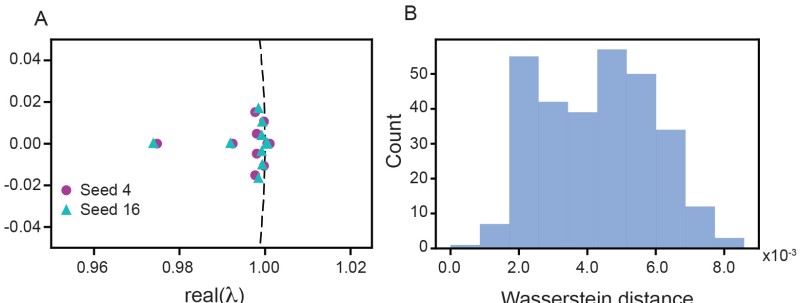

Figure S4: **Conjugate training dynamics across random initializations of FCNs.** (A) Example Koopman spectra associated with training FCNs, $h = 40$, from two different random seeds. (B) Distribution of Wasserstein distance between the Koopman spectra of all possible pairs of 25 independently trained FCNs.

permutation symmetry was not taken into account [16]. Our $h = 40$ FCNs strike a balance between these two points, making it a good point for analysis.

We find examples of different random initializations with nearly identical Koopman spectra (Fig. S4A). Across all possible pairs of the 25 independently trained FCNs, we find that the Wasserstein distance between Koopman spectra is centered around values similar to what was seen when comparing $h = 10$ and $h = 40$ FCNs. These results support the hypothesis that, when taking into account permutation symmetry (which computation of the Koopman eigenvalues naturally does), at least some random initializations have conjugate training dynamics.

## D    Convolutional neural network training phases

### D.1    Training experiment details

LeNet [60] and ResNet-20 [61] models were trained on MNIST and CIFAR-10 (respectively), for 20 epochs. Training was performed using the ShrinkBench framework [74], which makes use of PyTorch. The open source code can be found here: `https://github.com/JJGO/shrinkbench/tree/master`. All hyper-parameters used for training are presented in Table S2. They were chosen to be the same between the two architectures to make the comparison between the two fair and were selected to match the hyper-parameters that were previously used to study LeNet's training dynamics [11]. Three independent seeds were trained ($n_n = 3$), each of which was initialized from 10 perturbed initializations ($n_s = 10$). All experiments were run on a MacBook Air with an Apple M1 chip, 1 CPU, and no GPUs. Code implementing our experiments can be found at `https://github.com/william-redman/Identifying_Equivalent_Training_Dynamics`.

### D.2    Computing Koopman mode decomposition

As with the FCNs (Sec. C.2), the observables used to construct the Koopman operator were time-delays of the weights going from the last hidden layer to the output layer. Eight time-delays, $d = 8$, were used and the top 10 Koopman modes were considered, based on a reduced SVD. The appropriateness of this was again verified by examining the amount of variance captured by the first 10 principal components.

| Hyper-parameters | Value |
|---|---|
| Learning rate ($\eta$) | 0.0012 |
| Batch size ($b$) | 60 |
| Optimizer | Adam |
| Epochs | 20 |
| Activation function | ReLU |

Table S2: **Hyper-parameters used for CNN training in Sec. 4.3.**

Figure S5: **Koopman-based framework enables identification of transitions in dynamics across the training of LeNet and ResNet-20.** (A)–(B) Same as Fig. 4A–B, but for dynamics computed over individual epochs.

### D.3 Evaluating CNN training phases at a coarser timescale

In addition to studying the earliest part of CNN training, which was the focus of previous work [27], we examined how CNN training dynamics evolved across the first 20 epochs of training. To reduce the computational cost associated with computing and comparing Koopman spectra, we computed the Koopman mode decomposition across all training iterations within a single epoch.

At this coarser timescale, we find that LeNet, trained on MNIST, and ResNet-20, trained on CIFAR-10, exhibit very similar evolutions in training dynamics (Fig. S5). Indeed, both see a continual reduction in Wasserstein distance between neighboring epochs as training time goes on. Interestingly, in both cases, there is a growing block diagonal of dark blue, that becomes especially strong at training epoch 13. This supports the general similarity in the evolution of training dynamics between different CNN architectures, as was previously observed [27].

## E Transformer grokking

### E.1 Training experiment details

Single hidden layer Transformers, with four attention heads, were trained on modular arithmetic using open source code [62]: `https://github.com/KindXiaoming/Omnigrok/tree/main`. As noted in the repository, this is a modified version of previously developed code [29]. We keep all hyper-parameters the same. Therefore, we refer the interested reader to the details presented in the repository and related papers. 20 independent seeds were trained ($n_n = 20$), each of which was initialized from 10 perturbed initializations ($n_s = 10$). All experiments were run on a MacBook Air with an Apple M1 chip, 1 CPU, and no GPUs. Code implementing our experiments can be found at `https://github.com/william-redman/Identifying_Equivalent_Training_Dynamics`.

### E.2 Computing Koopman mode decomposition

As with the FCNs (Appendix C.2) and CNNs (Appendix D.2), we use as observables time-delays of the weights from the last hidden layer to the output. $d = 32$ time-delays were used. Because the number of weights was $> 65000$, before performing the time-delays and flattening the tensor that stored all the weights, we performed dimensionality reduction. This was achieved by applying PCA to the flattened tensor that contained all weights (without time-delays), and then projecting the weight trajectories corresponding to the training of each perturbed initialization onto the top 10 principal components. These dimensionally reduced weights were then time-delayed and used to construct the KMD. The top 10 Koopman modes were considered, based on a reduced SVD. The

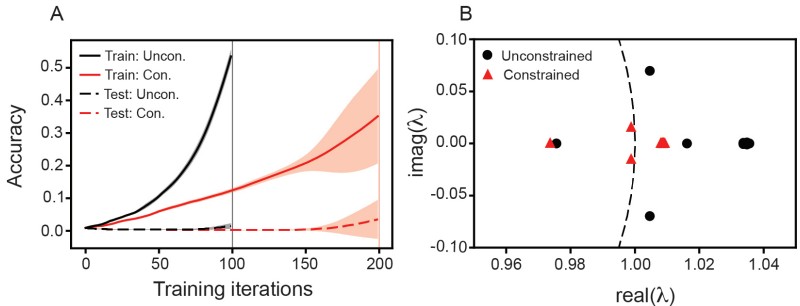

Figure S6: **Distinct Koopman eigenvalues between Transformers that do and do not undergo grokking is consistent when considering windows of training time that have more similar training loss.** (A) As in Fig. 5C, we train Transformers with unconstrained weight norm for $T = 100$ (thin vertical black line) iterations and use the associated weight trajectories to approximate the KMD. In contrast, here we train Transformers with constrained weight norm for $T = 200$ iterations (thin vertical red line) and use the associated weight trajectories to approximate the KMD. This enables the two networks to reach more comparable (although not perfectly matching) training losses. Lines are mean across 20 indepedently trained Transformers and shaded area is $\pm$ one standard deviation. (B) Same as Fig. 5D, but with the Koopman eigenvalues associated with training the constrained Transformer for $T = 200$ iterations.

appropriateness of this was again verified by examining the amount of variance captured by the first 10 principal components.

We found in Sec. 4.4 that the Transformers trained with constrained weight norm (that do not undergo grokking) have non-conjugate dynamics with the Transformers trained with unconstrained weight norm (that do undergo grokking). To ensure that this was not due to the fact that training loss over the window of training time used for computing the Koopman eigenvalues (the first $T = 100$ training iterations) was distinct between the two Transformers, we performed the following control experiment. Namely, we performed the same analysis, but we considered the weight trajectories of Transformers with constrained weight norm over twice as long a training time interval ($T = 200$). In this case, the constrained Transformer reaches a training loss that is closer to that of the unconstrained Transformer (Fig. S6A – compare solid red and black lines), although there is more variability between independent seeds (Fig. S6A – shaded red area). However, in this case we again find that the Koopman eigenvalues associated with unconstrained and constrained training are distinct (Fig. S6B). In particular, the constrained Transformer again has a pair of complex conjugate Koopman eigenvalues that lie along the unit circle, while the unconstrained Transformer has a pair of complex conjugate Koopman eigenvalues outside of the unit circle. This suggests distinct stability properties, and further emphasizes the absence of a conjugacy.

