# OpenReview forum: "Identifying Equivalent Training Dynamics"
_NeurIPS.cc/2024/Conference — NeurIPS 2024 spotlight_

### Official Review · Reviewer_4cZ9 · 2024-07-10

**Soundness:** 3
**Presentation:** 4
**Contribution:** 3
**Rating:** 8
**Confidence:** 3

**Summary:**

This paper proposes a method for identifying equivalent training dynamics of deep neural networks from the perspective of dynamical systems theory, in particular the spectral analysis of Koopman operators. The authors propose to use the notion of topological conjugacy, i.e. two dynamical systems are considered equivalent if a smooth invertible map can transform the trajectories of one system into trajectories of the other. Since in case of linear systems topological conjugacy holds if and only if the eigenvalues of the two (linear) systems are identical, the authors propose to utilize Koopman operator theory which precisely concerns linearization and eigendecomposition of non-linear dynamical systems, along with an advantage in the context of analyzing training dynamics of neural networks as Koopman eigenvalues (eigenvalues of a linearized dynamical system given by eigendecomposition of Koopman operator) are invariant up to permutation of state variables, i.e. they automatically account for the permutation symmetries of neurons (weights). Using a variant of dynamic mode decomposition (DMD) algorithm for approximately obtaining Koopman eigenvalues, and by comparing the sets of eigenvalues obtained from different training trajectories using Wasserstein distance, the authors report the following results:

- The proposed method recovers a recently discovered nonlinear topological conjugacy between the optimization dynamics of online mirror descent and online gradient descent on two synthetic regression problems,
- The proposed method identifies a fundamental change of training dynamics as the width of a MLP with a single hidden layer and ReLU activation function is increased from 5 to 10, in comparison to an increase from 10 to 40 which is less significant in terms of the difference in the eigenvalue spectrum.
- The proposed method identifies transitions in dynamics during early training of LeNet and ResNet-20 convolutional neural networks, and non-conjugacy of training dynamics between LeNet and ResNet-20 overall.
- The proposed method identifies non-conjugacy of training dynamics for transformers that do and do not exhibit grokking (delayed perfect generalization on small algorithmic datasets) controlled by constraining weight norm.

**Strengths:**

- S1. The paper is very well written and easy to follow.
- S2. The use of topological conjugacy to define equivalence of training dynamics rather than trajectories of parameters or loss is convincing, and original as far as I am aware; the most related work I could find was Bielecki (2002) which studies topological conjugacy of a system under a gradient flow and its Euler discretization.
- S3. The proposed method of estimating the eigenvalues of the Koopman operator, using a variant of DMD algorithm, and comparing them with Wasserstein distance, are natural and seems technically sound in implementing the idea of detecting topological conjugacy.
- S4. The fact that Koopman eigenvalues are invariant to permutation of neurons (weights) is quite useful, especially given the role of permutation symmetries in training dynamics of neural networks (Entezari et al. 2021; Bokman and Kahl, 2023).
- S5. The experimental results support the proposed idea and method, recovering a number of known results on equivalence of dynamics and falsifying some of the previously argued equivalences (e.g. of convolutional networks after the early transition of transition dynamics).

Bielecki, Topological conjugacy of discrete time-map and Euler discrete dynamical systems generated by a gradient flow on a two-dimensional compact manifold (2002)

Entezari et al. The role of permutation invariance in linear mode connectivity of neural networks (2021)

Bokman and Kahl, Investigating how ReLU-networks encode symmetries (2023)

**Weaknesses:**

- W1. There are some standard assumptions used in Koopman operator theory that may not strictly hold for mini-batch stochastic gradient descent, which includes non-stochasticity, forward-completeness, and time-invariance (violated e.g. due to learning rate scheduling and the use of momentum-based optimizers). Addressing these points may strengthen the paper.
- W2. While most of the presented experiments on neural networks find non-conjugacy of training dynamics, it would be impactful if the proposed method identifies *conjugacy* of seemingly non-equivalent training dynamics, e.g. of differently initialized neural networks as conjectured by Entezari et al. (2021).
- W3. The notations used in Equation (5) may be not exact; if we let $\rho\_\sigma:x\mapsto\tilde{x}$, it seems we might need to have $U^t \tilde{g}(\tilde{x})=\sum\_{i=1}^N\lambda\_i^t\tilde{\phi}(\tilde{x})v_i$ where $\tilde{g}\coloneqq g\circ \rho\_\sigma^{-1}$ and $\tilde{\phi}\coloneqq \phi\circ \rho\_\sigma^{-1}$. This will still leave Koopman eigenvalues invariant. (Please correct me if I'm wrong.)
- W4. It may be beneficial to add a non-conjugate optimization algorithm to Figure 2 to show that the similarity of spectra of online mirror descent and online gradient descent is indeed significant.
- W5. The use of Kolmogorov-Smirnov test in the panel F of Figure 3 seemed a bit weird since it shows that the difference of spectra of h=10 and h=40 is less than the difference of h=5 and h=10/40, but it does not precisely show the conjugacy of h=10 and h=40 as argued in Line 187-188; what would precisely show the conjugacy is a very low Wasserstein distance close to 0 (or a Wasserstein distance similar to random chancel level e.g. obtained from randomized partitioning of the eigenvalues of h=10 and h=40).

Minor comments and typos
- In the caption of Figure 6, should "... that are conjugate" be "... that are non-conjugate"?

Entezari et al. The role of permutation invariance in linear mode connectivity of neural networks (2021)

**Questions:**

- Q1. While the authors point out invariance to permutation symmetries of neurons (weights) in Equation (5), it seems that Koopman eigenvalues should be invariant for arbitrary invertible transformation symmetries on the parameters, which includes but not limited to permutations. Namely, rescaling symmetries (of cascaded linear layers) and rotation symmetries (of query and key projections used in attention in transformers) are other examples of invertible symmetries on the parameters (Ziyin, 2023). Please correct me if I am wrong.

Liu Ziyin, Symmetry induces structure and constraint of learning (2023)

**Limitations:**

The authors have discussed the limitations in Section 5.

---

> ### Author Rebuttal · Authors · 2024-08-06
>
> We thank the reviewer for their time and detailed comments. We are encouraged that they found our work well written and our framework novel! Below, we respond to the specific questions the reviewer had.
>
> **"It would be impactful if the proposed method identifies conjugacy of seemingly non-equivalent training dynamics"**
>
> This is a great suggestion. To address this, and to put it in the context of Entezari et al. (2021), we have analyzed the Koopman eigenvalues associated with training the $h = 40$ FCN on MNIST, across 25 initializations. We computed the Wasserstein distance between all pairs of eigenvalues and found that the distribution has a similar mean as what is seen in comparing $h = 10$ and $h = 40$ (compare Fig. 2B in the global response pdf with Fig. 3F in our submission). In addition, we find examples where the eigenvalues are nearly exactly overlapping (Fig. 2A in the global response pdf). This provides evidence that our framework can identify similar dynamics for FCNs that converge to solutions previously shown to be separated by a loss barrier [Fig. 2 left of Entezari et a. (2021)]. We will add these results to the main text in our revised manuscript (Sec. 4.2). We appreciate this suggestion.
>
> **"It may be beneficial to add a non-conjugate optimization algorithm to Figure 2 to show that the similarity of spectra of online mirror descent and online gradient descent is indeed significant"**
>
> This is a great idea and we believe that the inclusion of a non-conjugate optimization algorithm will additionally further emphasize how training dynamics can be different. To make this point most clear, we have chosen to compare online mirror and online gradient descent with the bisection method (BM). Unlike OMD and OGD, which take small local steps, the BM takes large, in some cases global, steps. We illustrate one trajectory in Fig. 3A of the global response pdf where these global steps lead to parameters that "hop" between positive and negative values (light blue lines). As such, we can intuitively expect that there is not a conjugacy between BM and OMD/OGD. Comparing the Koopman eigenvalues associated with each optimizer, we indeed see a clear difference between BM and OMD/OGD. In particular, BM has eigenvalues that are complex and negative, which correspond to the "hopping" in the parameter space. In contrast, OMD/OGD have positive, real-only Koopman eigenvalues. This further emphasizes what differences in Koopman eigenvalues correspond to. We will add these results to the main text in our revised manuscript (Sec. 4.1). We appreciate this suggestion.
>
> **"While the authors point out invariance to permutation symmetries of neurons (weights) in Equation (5), it seems that Koopman eigenvalues should be invariant for arbitrary invertible transformation symmetries on the parameters, which includes but not limited to permutations"**
>
> This is a great comment. Thank you for pointing this out. We were focused on the permutation symmetry, but you are correct that there are other symmetries that are relevant in the DNN training setting that the Koopman spectra are invariant to. We will discuss this in the Introduction and Sec. 3 of the revised manuscript. We believe that this further demonstrates the utility of our framework.
>
> **"There are some standard assumptions used in Koopman operator theory that may not strictly hold for mini-batch stochastic gradient descent"**
>
> This is a great point. We agree that more discussion on why we think that Koopman mode decomposition can be reasonably approximated in these settings, and what we believe can be done to improve the approximation, would strengthen our paper. Additionally, it would make it more clear what challenges might be expected in utilizing our framework to study other problems. We will add this in Sec. 5 of the revised manuscript.
>
> We note that in the FCN setting, no learning rate scheduling or momentum-based optimization was used, thus making it most directly in-line with previous work that has used dynamic mode decomposition with time-delays to study noisy dynamical systems [Arbabi and Mezic (2017); Brunton et al. (2017)].
>
> **"The notations used in Equation (5) may be not exact"**
>
> Yes! Thank you for that correction. We will correct this in the revised version of the manuscript.
>
> **"The use of Kolmogorov-Smirnov test in the panel F of Figure 3 seemed a bit weird since it shows that the difference of spectra of h=10 and h=40 is less than the difference of h=5 and h=10/40, but it does not precisely show the conjugacy of h=10 and h=40"**
>
> This is a good point and we agree that in general it is challenging to show conjugate dynamics with only the Wasserstein distance. For this reason, we explicitly examined the Koopman spectra of $h = 5$, $10$, and $40$ (Fig. 3D and E in submission). We find that the Koopman eigenvalues between $h = 10$ and $h = 40$ are not only close to each other, but have similar properties (i.e., if an eigenvalue for $h = 40$ is real only, the eigenvalue of $h = 10$ that is closest to it is also real only). This is not the case for $h = 5$, where there is a pair of complex conjugate eigenvalues that do not match the real only eigenvalues of $h = 10$/$40$. For this reason, we note that this “suggests conjugate dynamical behavior” (lines 187-188). In the revised manuscript, we will make it more clear why we are concluding this, and that this conclusion is a hypothesis, given the current tools we have.
>
> **"In the caption of Figure 6, should "... that are conjugate" be "... that are non-conjugate"? "**
>
> Oops yes! Thank you for pointing that out. We have corrected that typo.
>
> **References:**
>
> Arbabi and Mezic (2017) “Ergodic theory, dynamic mode decomposition, and computation of spectral properties of the Koopman operator”
>
> Brunton et al., (2017)  “Chaos as an intermittently forced linear system”
>
> Entezari et al. (2021) "The role of permutation invariance in linear mode connectivity of neural networks"

---

> ### Comment · Reviewer_4cZ9 · 2024-08-13
>
> Thank you for the comprehensive response. The added experiments and discussions would greatly improve the impact of the paper and I have raised my score accordingly, and my only remaining concern is on the proper establishment of the null for at least one experiment, as in line with the request of reviewer 42wo. This seems important as the added results also use Wasserstein distances to show the conjugacy of differently initialized FCNs. If this is addressed, I am willing to raise my score to 8.
>
> Sorry for requesting this close to the end of the discussion period, but I think this may be feasible since the random permutation baseline (take two sets of eigenvalues, measure their Wasserstein distance, then permute them, then measure their Wasserstein distance, then compare these two distances) does not require running the training again.

---

> > ### Comment · Reviewer_4cZ9 · 2024-08-13
> >
> > I have checked the response to reviewer 42wo and have further updated my score. I wish the authors the best of luck.

---

> > > ### Author Response · Authors · 2024-08-13
> > >
> > > We thank the reviewer for their time and willingness to engage with our work!

---

### Official Review · Reviewer_ZFpv · 2024-07-12

**Soundness:** 3
**Presentation:** 3
**Contribution:** 3
**Rating:** 6
**Confidence:** 3

**Summary:**

To compare two training dynamics, the paper proposes to compare Koopman operators for them, especially their eigenvalues, based on the previous result showing that consensus in the Koopman eigenvalues implies the equivalence of two dynamics up to some homeomorphism. Their framework is applied to compare various neural network architectures, whose results provide fine-grained insights to the previous (ad-hoc) findings.

**Strengths:**

- The paper is clearly written, and well-structured.
- The proposed framework seems natural, and can be applicable to various training dynamics.

**Weaknesses:**

- There remain a concern about the computational efficiency of computing Koopman eigenvalues in the proposed framework, especially when the parameter space is high-dimensional.
- The framework may be not suitable to compare the dynamics of different dimensions because the equality of the dimensions is required for the homeomorphic equivalence, according to basic results in topology.

**Questions:**

See weaknesses.

**Limitations:**

Yes

---

> ### Author Rebuttal · Authors · 2024-08-05
>
> We thank the reviewer for their time and helpful comments. We are encouraged that they found our work well written and our framework applicable. Below, we respond to the specific questions the reviewer had.
>
> **"There remains a concern about the computational efficiency of computing Koopman eigenvalues in the proposed framework, especially when the parameter space is high-dimensional."**
>
> We thank the reviewer for pointing out that this limitation was not addressed in our submission. We agree that computing the Koopman eigenvalues can be a computationally intensive procedure. However, there are several ways of mitigating this. First, for especially large parameter spaces, dimensionality reduction (e.g., via PCA) can be used to reduce the size of the matrices used to approximate the Koopman mode decomposition, without sacrificing much dynamical information. Second, instead of considering all the Koopman eigenvalues, the top-$n$ can be considered (where small $n$ can often be sufficient and can reduce the noise present in the dynamics). This enables a reduced SVD, which can aid in reducing the computational costs. Third, a subset of the total number of parameters can be chosen as observables. This is motivated by the fact that in many DNNs there are a small number of highly correlated sets of weights [Brokman et al. (2023)], and thus, only a small number of the total parameters is needed to get an accurate picture of the dynamics. And fourth, numerical implementations of Koopman mode decomposition have been integrated into LAPACK [Drmac (2024)] (one of the most widely integrated libraries for numerical linear algebra), and are being developed for high performance computing. Therefore, tools exist to examine even larger parameter spaces. We will discuss this limitation in the Sec. 5 of the revised manuscript.
>
> **"The framework may be not suitable to compare the dynamics of different dimensions"**
>
> This is a good point that we did not address in our submission. In dynamical systems theory, in addition to the notion of a topological conjugacy, there is the notion of a semi-conjugacy which is defined for systems of different dimensions. In this case, the semi-conjugacy $h$ is a smooth but non-invertible mapping satisfying Eq. 1. Semi-conjugacies can be identified by one dynamical system having Koopman eigenvalues that are a subset of another dynamical system. We will discuss semi-conjugacies in Sec. 3.1 and 3.3 of our revised manuscript.
>
> **References:**
>
> Brokman et al. (2023) "ENHANCING NEURAL TRAINING VIA A CORRELATED DYNAMICS MODEL"
>
> Drmac (2024) "A LAPACK Implementation of the Dynamic Mode Decomposition"

---

> > ### Comment · Reviewer_ZFpv · 2024-08-12
> >
> > Thank you for the additional clarifications. I would like to keep my score.

---

### Official Review · Reviewer_42wo · 2024-07-21

**Soundness:** 3
**Presentation:** 3
**Contribution:** 4
**Rating:** 7
**Confidence:** 4

**Summary:**

This paper applies Koopman mode decomposition (KMD) to examine whether the training dynamics of different model architectures/optimizers are equivalent or not. In particular the authors examine 4 cases: 1) the same objective learned by online mirror descent vs online gradient descent (which are equivalent), 2) the effect of width on similarity between FCNs with one hidden layer trained on MNIST, 3) training dynamics and especially the early phase of CNNs (known to rapidly evolve and then settle into a consistent phase), and 4) grokking in Transformers (known to have delayed test improvement if weight norms are unconstrained).

**Strengths:**

The use of Koopman mode decomposition for analyzing training dynamics is novel as far as I can tell, and addresses important questions about neural network similarity, such as whether networks are equivalent after eliminating the effect of certain symmetries. These types of questions are difficult to answer and currently require specific methods to target specific symmetries. Having a general method that accounts for many symmetries is obviously advantageous, as is having a method that looks directly at the time evolution of models rather than their various outputs. The paper succinctly presents an easy to follow overview of Koopman operators and their applications with the necessary citations.

**Weaknesses:**

Unfortunately, the results lack points of comparison and do not provide sufficiently novel or precise interpretations.

The main issue is what would constitute a "low" vs. "high" Wasserstein distance between the Koopman eigenvalues? The first task examines similarity and the remainder dissimilarity, but all tasks need baselines for both known similarities and known dissimilarities between models. Otherwise, it is impossible to decide on a threshold between the two interpretations. This is similar to the issues with interpreting representational similarity metrics (e.g. Kornblith et al. Similarity of Neural Network Representations Revisited) and one may look at works in that field for an idea of how to make experiments more interpretable.

It is hard to conceptually understand what similar versus distinct training dynamics look like. Some more toy examples like task 1, perhaps with distinct minima, could be helpful in this regard.

The interpretations are all confirming previously known results, so it would be helpful to present a new result or refine a previously know result, that can also be confirmed by other means. For example, the method could be used to predict linear mode connectivity onset on a class of models/optimizers that has not been previously examined by Frankle et al. (using their empirical methods to confirm the onset time). This would establish that the method has real interpretive power.

**Questions:**

Why not examine loss/error barriers instead of loss (as in Frankle et al.) to determine if the perturbed networks in appendix B.2 have similar trajectories?

What is the significance of keeping the product of time delay and width constant? How does the frequency of time samples affect the estimated eigenvalues? How about noise (e.g. from SGD)?

Would sampling Bayesian neural networks allow for arbitrarily many weights from the same training dynamic?

How does this method allow grokking to be identified early?

**Limitations:**

A major limitation of this method is that it requires many runs of a model to give good estimates of the Koopman eigenvalues. This makes designing experiments both more tricky and computationally expensive. Specifically, the difficulty is in having different runs that are known to have similar training dynamics, yet different weights. I'm this work, random multiplicative Gaussian perturbations are used. But there seems to be no way to directly check that these runs are actually similar in training dynamics. One baseline might be to evaluate KMD on independent sets of the same kind of run, and see how much the eigenvalues vary.

The authors mention that eigenvalues greater than 1 correspond to unstable dynamics. It would be interesting to elaborate on what this means and how it relates to other observations or settings.

---

> ### Author Rebuttal · Authors · 2024-08-06
>
> We thank the reviewer for their time and detailed comments. We are encouraged that they found our work well written and our framework novel. Below, we respond to the specific questions the reviewer had.
>
> **"The main issue is what would constitute a "low" vs. "high" Wasserstein distance between the Koopman eigenvalues?"**
>
> We appreciate the reviewer pointing out this limitation, as it brings to our attention that we can strengthen our explanation of this in the manuscript. We agree that there is, at the moment, no single number that enables us to identify a high Wasserstein distance vs. a low Wasserstein distance. However, we believe that there are several features that make our framework novel and interpretable.
>
> First, it is worth noting that there existed no principled way of studying training dynamics (indeed, this is what motivated us). This has hindered prior work attempting to study training dynamics. For instance, Frankle et al. (2020) plotted several different measurements of training dynamics (e.g., loss, weight trajectories, weight norms) to identify transitions. We believe this is less interpretable and rigorous than our method.
>
> Second, the Koopman eigenvalues can be understood as corresponding to different time-scales of the dynamics. When different features of the Koopman eigenvalues are present (e.g., real-only Koopman eigenvalues, complex conjugate Koopman eigenvalue pairs, Koopman eigenvalues with magnitude > 1), this corresponds to distinct properties of the dynamics (e.g., exponential decay, oscillations, exponential growth). Comparison of the individual Koopman eigenvalues can aid in the interpretation of the results. For instance, in the case of the fully connected neural networks, the wider networks have a Koopman eigenvalue that is real only, while the narrowest network has a Koopman eigenvalue complex conjugate pair. This, in addition to the large Wasserstein distance, led us to conclude that there was a non-conjugacy. This is a point that we will make more clear in the revised version of the manuscript by adding text to Secs. 3.2 and 4.1-4.4.
>
> **"It is hard to conceptually understand what similar versus distinct training dynamics look like."**
>
> We agree that making the distinction between conjugate and non-conjugate training dynamics intuitive is important to effectively communicating our method. To address this, we considered the bisection method (BM), an optimizer that is not conjugate to online mirror and gradient descent. Unlike OMD and OGD, which take small local steps, the BM takes large, in some cases global, steps. We illustrate one trajectory in Fig. 3A of the global response pdf where these global steps lead to parameters that "hop" between positive and negative values (light blue lines). As such, we can intuitively expect that there is not a conjugacy between BM and OMD/OGD. Comparing the Koopman eigenvalues associated with each optimizer, we indeed see a clear difference between BM and OMD/OGD. In particular, BM has eigenvalues that are complex and negative, which correspond to the "hopping" in the parameter space. In contrast, OMD/OGD have positive, real-only Koopman eigenvalues. This further emphasizes what differences in Koopman eigenvalues correspond to. We will add these results to the main text in our revised manuscript (Sec. 4.1).
>
> **"The interpretations are all confirming previously known results, so it would be helpful to present a new result or refine a previously known result"**
>
> We would like to emphasize that, while all our experiments were motivated by previous work, our framework enabled us to obtain novel results. Namely, we showed that LeNet and ResNet-18 have different training dynamics (while having similar transitions in their dynamics) and that Transformers that do and do not undergo grokking exhibit distinct training dynamics. We see these as being illustrative of the power of our method. Additionally, inspired by the comments of reviewer 4cZ9, we examined whether our framework could identify conjugate training dynamics in FCNs that have different initializations, as hypothesized by Entezari et al. (2021). As shown in Fig. 2 of the global response pdf, we find evidence in support of this hypothesis. This is another novel result. We will add it to the main text of the revised manuscript (Sec. 4.2).
>
> **"Why not examine loss/error barriers instead of loss (as in Frankle et al.) to determine if the perturbed networks in appendix B.2 have similar trajectories?"**
>
> This is a great idea! We will incorporate this into the revised version of the manuscript. We appreciate the reviewer suggesting this.
>
> **"How does this method allow grokking to be identified early?"**
>
> The hypothesis that our method may enable grokking to be identified early in training comes from the fact that, when there are Koopman eigenvalues with magnitude > 1, there is an instability in the dynamics of the system. To see this, note that in the Koopman mode decomposition, the Koopman eigenvalues are raised to powers of the number of time steps in the future (Eq. 4). Thus, if an eigenvalue has magnitude > 1, its mode will quickly dominate the decomposition, and then explode. In the context of DNN training, this could be indicative of either exploding gradients, or a short-term transition in the training dynamics that, over a brief time window, appears to be an instability. That we find Transformers that do grok to have more eigenvalues > 1 suggests a greater instability in the training dynamics. Therefore, we hypothesized that looking for these eigenvalues may enable the identification of Transformers that will undergo grokking ahead of time. We appreciate the reviewer pointing out that this was not sufficiently explained in our submission and we will add text to clarify this in Secs. 3.2 and 4.4.
>
> **References:**
>
> Frankle et al. (2020) "The Early Phase of Neural Network Training"

---

> > ### Comment · Reviewer_42wo · 2024-08-07
> >
> > Thank you for the detailed response and additional figures.
> >
> > The discussion of how to interpret the eigenvalues, especially for grokking, has convinced me of this method's utility and impact. The BM versus OMD/OGD experiment (figure 3) and the confirmation of Entezari et al.'s conjecture (figure 2) also convincingly demonstrate the method's ability to identify both positive and negative results.
> >
> > I agree with reviewer ZFpv on the following:
> > > W5 [...] what would precisely show the conjugacy is a very low Wasserstein distance close to 0 (or a Wasserstein distance similar to random chancel level e.g. obtained from randomized partitioning of the eigenvalues of h=10 and h=40).
> >
> > This is in line with my request for a baseline Wasserstein distance among equivalent models. Either permutation testing (as suggested by reviewer ZFpv), or establishing a null distribution for distance between independently sampled sets of models (as suggested by myself) would work in this regard. I will readily recommend acceptance if baselines are established for the distances reported in figures 3-5, and optionally 6 as well (specifically, a threshold of significance would clearly indicate when we can consider two models to have equivalent training dynamics).
> >
> > Finally, could the authors comment on two of my questions below?
> > - What is the significance of keeping the product of time delay and width constant? How does the frequency of time samples affect the estimated eigenvalues?
> > - Would sampling Bayesian neural networks allow for arbitrarily many weights from the same training dynamic?

---

> > > ### Author Response · Authors · 2024-08-08
> > >
> > > We thank the reviewer for their quick response! We are glad to hear that the new experiments and explanations were helpful in better demonstrating our work's utility/impact.
> > >
> > > We appreciate the reviewer bringing up reviewer ZFpv's idea on creating a baseline comparison by sampling across eigenvalues from different architectures. Upon thinking about this more, we agree that this is a good idea and an important metric to include for identifying conjugacies. We will implement this and include it in our analysis of Figs. 2-6 in our revised manuscript.
> > >
> > > And yes! We apologize for not addressing those questions earlier. We had written responses, but ran out of the rebuttal's 6000 character limit. Our responses are below.
> > >
> > > **"What is the significance of keeping the product of time delay and width constant? How does the frequency of time samples affect the estimated eigenvalues?"**
> > >
> > > This is a good question and we appreciate the reviewer pointing out that this was not clear in our submission. We kept the ratio of time delays to width constant simply to enable a more fair comparison between the architectures. Not keeping the ratio fixed would lead to more observables for wider networks than narrower networks, as there are more weights in the former case than the latter case. This could artificially affect the comparison of Koopman eigenvalues. We will add discussion of this to the main text of our revised manuscript (Sec. 4.2).
> > >
> > > With regards to frequency of sampling, it is important to keep this consistent across comparisons of different networks, as different samplings can lead to different Koopman eigenvalues. For instance, a sparser sampling will make it appear that the weights change more than a denser sampling (as the time step is larger for the former than the latter case). However, differences in sampling should not affect the stability properties of the Koopman eigenvalues (> 1 or < 1). We will add discussion of this to the main text of our revised manuscript (Sec. 3.3).
> > >
> > > **"Would sampling Bayesian neural networks allow for arbitrarily many weights from the same training dynamic?"**
> > >
> > > We are unfamiliar with Bayesian neural networks and are uncertain of what exactly is being asked here. Is the idea to use the Koopman eigenvalues to generate networks with specific training trajectories? If so, this would be an interesting extension of this work. This would also be similar to (but build upon) previous work by Dogra and Redman (2020), Tano et al. (2020), and Luo et al. (2023) who used Koopman mode decomposition to evolve neural networks forward in time in a data-independent manner.
> > >
> > > **References:**
> > >
> > > Dogra and Redman (2020) "Optimizing neural networks via Koopman operator theory"
> > >
> > > Tano et al. (2020) "Accelerating Training in Artificial Neural Networks with Dynamic Mode Decomposition"
> > >
> > > Luo et al. (2023) "QuACK: Accelerating Gradient-Based Quantum Optimization with Koopman Operator Learning"

---

> > > > ### Comment · Reviewer_42wo · 2024-08-12
> > > >
> > > > > We appreciate the reviewer bringing up reviewer ZFpv's idea on creating a baseline comparison by sampling across eigenvalues from different architectures. Upon thinking about this more, we agree that this is a good idea and an important metric to include for identifying conjugacies. We will implement this and include it in our analysis of Figs. 2-6 in our revised manuscript.
> > > >
> > > > Apologies for the short notice, but would it be possible to give an idea of the magnitude of the baseline relative to the results in the paper, at least for a subset of the experiments?
> > > >
> > > > > We kept the ratio of time delays to width constant simply to enable a more fair comparison between the architectures. Not keeping the ratio fixed would lead to more observables for wider networks than narrower networks, as there are more weights in the former case than the latter case. [...] With regards to frequency of sampling, it is important to keep this consistent across comparisons of different networks, as different samplings can lead to different Koopman eigenvalues.
> > > >
> > > > I am a bit confused by this, in that as far as I understand the wider architectures were sampled less frequently in your work, but this could then change the Koopman eigenvalues as you indicated. Would it be possible to sample more independent runs of the narrower models instead, while keeping the sampling frequency constant for all architectures?
> > > >
> > > > > We are unfamiliar with Bayesian neural networks and are uncertain of what exactly is being asked here. Is the idea to use the Koopman eigenvalues to generate networks with specific training trajectories? [...]
> > > >
> > > > Sorry for not being clear. What I meant was instead of using different independent runs to get enough weight data, would it be possible to train one Bayesian neural network and then sample weights from its posterior distribution to compute the Koopman eigenvalues? This could allow the method to capture training dynamics from a single model run. For example, training with stochastic gradient Langevin dynamics (Welling & Teh, Bayesian Learning via Stochastic Gradient Langevin Dynamics) essentially parameterizes the model with Gaussians. At a given checkpoint, one can then generate as many sets of weights as needed by sampling the Gaussian distribution at that time step.

---

> > > > > ### Author Response · Authors · 2024-08-13
> > > > >
> > > > > We thank the reviewer for their continued engagement in our submission! We appreciate their time and their efforts to improve our work.
> > > > >
> > > > > **"Would it be possible to give an idea of the magnitude of the baseline relative to the results in the paper, at least for a subset of the experiments?"**
> > > > >
> > > > > We are still working on getting all the baselines implemented, but we have run the comparison on the mirror descent, gradient descent, and bisection method example.
> > > > >
> > > > > To assess how distinct the Koopman eigenvalues are, we take the eigenvalues $\lambda_1$ and $\lambda_2$, associated with two of the different optimizers, and then perform a "shuffling", randomly assigning elements of $\lambda_1$ and $\lambda_2$ to $\lambda'_1$ and $\lambda'_2$. To ensure a fair comparison, we first sort the eigenvalues and then shuffle between the sorted values (so that $\lambda'_1$ isn't only large norm eigenvalues and $\lambda'_2$ isn't only small norm eigenvalues). Additionally, we ensure that complex conjugate pairs of eigenvalues are shuffled to the same set (either $\lambda'_1$ or $\lambda'_2$). We then compute the Wasserstein distance between $\lambda'_1$ and $\lambda'_2$. We perform 100 such shuffles and then compare the distribution of Wasserstein distances with the true Wasserstein distance.
> > > > >
> > > > > We find that for mirror descent and gradient descent (which are known to be conjugate), 26% of the shuffles have greater than or equal to Wasserstein distance as the true Wasserstein distance. This suggests that the true eigenvalues of mirror and gradient descent are not significantly different from each other. In contrast, 1% of the shuffles have a greater Wasserstein distance than the true Wasserstein distance, when comparing gradient descent and the bisection method (non-conjugate algorithms). Similarly, 2% of the shuffles have a greater Wasserstein distance than the true Wasserstein distance, when comparing mirror descent and the bisection method (non-conjugate algorithms). This suggests that the Koopman eigenvalues associated with mirror and gradient descent are significantly distinct from the eigenvalues of the bisection method.
> > > > >
> > > > > **"I am a bit confused by this, in that as far as I understand the wider architectures were sampled less frequently in your work, but this could then change the Koopman eigenvalues as you indicated."**
> > > > >
> > > > > We apologize for the confusion. By time-delays, we do not mean the sampling frequency, but instead an approach typically used to generate observables in dynamical systems theory (see Taken's Embedding on Wikipedia for a nice example). To be more clear, we sample the weights of the $h = 5$, $h = 10$, and $h = 40$ networks all at the same frequency. However, because there are more weights in the $h = 40$ case, we increase the number of observables we use to construct the Koopman operator for the $h = 5$ and $h = 10$ cases through using more time-delays. We will make this more clear in our revised manuscript (Sec. 3.2).
> > > > >
> > > > > **"What I meant was instead of using different independent runs to get enough weight data, would it be possible to train one Bayesian neural network and then sample weights from its posterior distribution to compute the Koopman eigenvalues?"**
> > > > >
> > > > > This is a great idea! We imagine this would definitely be more efficient and enable a more detailed understanding of the dynamics for a given network. We will discuss this future direction in the revised manuscript (Sec. 5)

---

> > > > > > ### Comment · Reviewer_42wo · 2024-08-13
> > > > > >
> > > > > > Hello, thank you for providing preliminary baselines and further replies addressing my questions. All of my concerns with the work have been addressed, and I have raised my score to reflect this. Thank you and good luck!

---

> > > > > > > ### Author Response · Authors · 2024-08-13
> > > > > > >
> > > > > > > Thank you!

---

### Official Review · Reviewer_C2X3 · 2024-07-21

**Soundness:** 4
**Presentation:** 3
**Contribution:** 4
**Rating:** 8
**Confidence:** 4

**Summary:**

The authors utilize topological conjugacy and Koopman operator theory to create a framework for identifying between conjugate and non-conjugate training dynamics in DNNs. To validate their approach, they first show that their framework can accurately identify the known equivalence between online mirror descent and online gradient descent optimization methods. They then use the framework to gain new insights into training dynamics across various DNN architectures:

- **Shallow vs. Wide Fully Connected Networks**: The framework reveals non-conjugate training dynamics between these types of networks.
- **CNNs**: The framework helps characterize the early training phase dynamics in CNNs.
- **Transformers**: The framework identifies non-conjugate training dynamics in Transformer models, including instances of "grokking".

Overall, the results demonstrate the framework's versatility and potential to illuminate the complex training dynamics of different DNN architectures.

**Strengths:**

- I very much enjoyed the paper. It tackles a significant and challenging problem in deep learning practice using a novel and precise analytical approach. Understanding training dynamics can enhance the efficiency and robustness of DNN models. Any effort to advance the understanding of training dynamics is highly valuable, and this paper provides excellent and helpful insights in this regard.

- Topological conjugacies have traditionally been difficult to compute. Using Koopman operator theory, the authors developed a novel framework for identifying conjugate and non-conjugate training dynamics.

- They investigated the training dynamics across various DNN architectures.

**Weaknesses:**

There are only a few concerns:

1- Regarding the identification of topological conjugacies using Koopman Operator Theory, what happens if the training dynamics are more complex, such as in the case of chaotic dynamics with a **mixed Koopman spectrum**? How can we investigate topological conjugacy in such situations?

2- It is still unclear to me whether and how a numerical method for approximating the KMD might affect the results of their method for identifying topological conjugacies.

3- The Wasserstein distance is used to quantify the differences between Koopman eigenvalues. Why was only this metric chosen to compare distributions? Could other metrics, such as Kullback-Leibler Divergence (KL Divergence) or Hellinger Distance, be used for comparison? Specifically, it would be nice to better understand more advantages of the Wasserstein distance over these other metrics. Additionally, could using multiple metrics (at least two different metrics) help assess the robustness of the results?

**Questions:**

I) On page 2, lines 59-60, the authors have mentioned that the same framework can be used across a number of DNN architectures to study a variety of dynamical phenomena during training demonstrates the generality of this approach. It would be helpful to understand how their approach could be applied to **RNNs for time series forecasting**. Could you please elaborate on this?

II) According to Table S1, the activation function for FCN is ReLU. I am interested in knowing how choosing another activation function (any activation function except ReLU) can affect the results?

**Limitations:**

The authors have adequately addressed most of the limitations. The other limitation that occurs to me is the limitations of their method for training dynamics that are more complex, such as in the case of chaotic dynamics with a mixed Koopman spectrum.

---

> ### Author Rebuttal · Authors · 2024-08-05
>
> We thank the reviewer for their time and detailed comments. We are encouraged that they found our work enjoyable to read and insightful! Below, we respond to the specific questions the reviewer had.
>
> **"How can we investigate topological conjugacy if the training dynamics are more complex, such as in the case of chaotic dynamics with a mixed Koopman spectrum?"**
>
> This is a great question! Recent work by Avila and Mezic (2024) has hypothesized that conjugacies in the case of continuous or mixed spectra can be identified by studying extensions of Koopman operator theory. We will explicitly discuss this in the revised manuscript. Additionally, we will make clear that systems with continuous spectra do not have guaranteed conjugacies if their point spectra match. However, we are unaware of any DNN training algorithms that would lead to chaotic trajectories (if you are familiar with any, we would be very interested in hearing!).
>
> **"I am interested in knowing how choosing another activation function (any activation function except ReLU) can affect the results?"**
>
> This is a good question and we appreciate the suggestion. We performed a new set of experiments, examining the effect of FCN width when using GeLU activation functions. We find very similar results to FCNs using ReLU activation functions (Fig. 1 of the global response pdf). This illustrates that our framework is robust to changes in activation function. We will add this figure to the Appendix and reference the results in Sec. 4.2 of the revised manuscript.
>
> **"Why was the Wasserstein distance used to quantify differences between Koopman spectra?"**
>
> We appreciate the reviewer pointing out that this choice was not well motivated in our submission. We chose the Wasserstein distance as it provides a metric of how far apart individual Koopman eigenvalues are from each other. This is in contrast to the KL divergence, which quantifies how different the distributions of eigenvalues are (but not whether the distributions could be easily made to be similar). The need for a metric that provides the distance between individual eigenvalues comes from the fact that the eigenvalues correspond to time-scales of the dynamics. Therefore, we expect two dynamical systems with Koopman eigenvalues that are far apart to have more distinct dynamics than two dynamical systems with Koopman eigenvalues that are near each other. We will add this rationale in Sec. 3.3 to make our choice more transparent.
>
> **"It is still unclear to me whether and how a numerical method for approximating the KMD might affect the results of their method for identifying topological conjugacies."**
>
> This is a good question that we hope future work will investigate in rigorous detail. However, we can provide some insight on this question.
>
> There are two ways that numerical methods for approximating the Koopman mode decomposition might impact the ability to properly identify topological conjugacies between DNN training dynamics. The first is if one set of DNN models has more noise in its training dynamics than another model. In this case, the approximated Koopman eigenvalues associated with the noisier training may not match the approximated Koopman eigenvalues associated with the less noisy training, even if a topological conjugacy between the average dynamical behavior exists. To mitigate this, we chose to use time-delays, which have been found to be more robust to noise [Arbabi and Mezic (2017); Brunton et al. (2017)], and we used a reduced SVD, so as to not compare Koopman eigenvalues associated with modes that are weaker, and thus more likely to be noise. Additionally, by sampling more than one trajectory, we reduced the impact an individual noisy training trajectory has.
>
> The second way numerical implementations of Koopman mode decomposition can impact the identification of topological conjugacies is if the numerical implementation is biased towards certain errors for specific dynamics. For instance, if the numerical implementation was biased towards generating complex conjugate pairs of eigenvalues when the dynamics have a fast exponential decay, but not if the dynamics have a slow exponential decay, this could lead to the conclusion of a greater difference in dynamics than what is actually present. However, we are unaware of any work showing that dynamic mode decomposition with time-delays (the numerical implementation we used) has this feature. Therefore, we do not believe it will have a major effect on our results.
>
> We will add this discussion to the main text of our revised manuscript (Sec. 5).
>
> **"It would be helpful to understand how their approach could be applied to RNNs for time series forecasting."**
>
> We had not thought about using our method in the context of RNNs for time-series forecasting, but our initial thoughts are that it could be leveraged in two ways. First, identifying that two seemingly distinct time-series induce conjugate training dynamics in a given RNN model would suggest that the time-series have some fundamental similarities. This could provide a unique way of doing systems identification and coarse-graining. And second, training RNNs can be challenging due to vanishing and exploding gradients. By computing the Koopman spectra associated with the early training of models that have these issues, and comparing them to ongoing training runs, it may be possible to identify ahead of time whether an RNN model undergoing training is likely to experience these problems. This could allow for more efficient early stopping and architecture search.
>
> **References:**
>
> Avila and Mezic (2024) “Spectral Properties of Pullback Operators on Vector Bundles of a Dynamical System”
>
> Arbabi and Mezic (2017) “Ergodic theory, dynamic mode decomposition, and computation of spectral properties of the Koopman operator”
>
> Brunton et al., (2017) “Chaos as an intermittently forced linear system”

---

> > ### Comment · Reviewer_C2X3 · 2024-08-13
> >
> > I appreciate the author's response and new experiments which have clarified my questions. I am satisfied with the responses, I have no further questions to discuss. So, I recommend acceptance of this paper.

---

### Author Rebuttal · Authors · 2024-08-06

We thank the reviewers for their time and thoughtful comments. A response to each individual reviewer’s comments is provided in the thread of the associated review. We believe that addressing these questions has greatly improved the quality of our work. Here, we highlight three new results that we obtained (see attached global response pdf):

**Robustness to different activation functions:** Motivated by reviewer C2X3, we trained fully connected networks of various widths on MNIST, using the GeLU activation function, instead of the ReLU activation function. We find very similar results when computing and comparing the Koopman eigenvalues (Fig. 1, global response pdf), illustrating that our framework is robust to changes in the activation function.

**Similar dynamics for FCNs with different initializations:** Motivated by reviewer 4cZ9, we examined whether our widest FCNs had similar Koopman eigenvalues, across different initializations and SGD seeds. This directly tests a hypothesis by Entezari et al. (2021) on different initializations having equivalent training dynamics, when taking into account the permutation symmetry (which the Koopman eigenvalues are invariant to). We find evidence supporting this hypothesis (Fig. 2, global response pdf), illustrating another example of when and how our framework can be used to address phenomena that have been challenging to address with existing methods.

**Distinct Koopman eigenvalues for non-conjugate optimizer:** Motivated by reviewer 4cZ9, we compared the Koopman spectra of online mirror and online gradient descent to a non-conjugate optimizer, the bisection method (BM). We find that OMD/OGD have significantly different Koopman eigenvalues than those associated with BM. In particular, because BM takes large, global steps that can lead to the changing of signs of the parameters being optimized, the Koopman eigenvalues are complex. This is in contrast with OMD/OGD, which takes local steps and has positive, real-only Koopman eigenvalues. This illustrates the significance of similarity in eigenvalues associated with OMD and OGD, and provides additional insight into what makes two training trajectories non-conjugate.

**References:**

Entezari et al. (2021) "The role of permutation invariance in linear mode connectivity of neural networks"

---

### Decision · Program_Chairs · 2024-09-25

**Decision:**

Accept (spotlight)

**Comment:**

The paper proposes a new method for identifying equivalent training dynamics of neural networks using Koopman operator theory and the notion of topological conjugacy. The paper received positive feedback overall: reviewers recognized the proposed analyzes of the training dynamics as novel, addressing important questions about neural network similarity and offering a general method that accounts for various symmetries; the potential impact of the method in understanding and improving neural network training dynamics was also acknowledged. The reviewers appreciated the clear presentation and found the method to be natural and applicable to various training dynamics. As per reviewers’ requests, the authors improved the paper by adding additional examples and experiments, and clarifying some of the assumptions.